 **eLIFE**

# Structural principles of SNARE complex recognition by the AAA+ protein NSF

K Ian White[1,2,3,4,5], Minglei Zhao[6], Ucheor B Choi[1,2,3,4,5], Richard A Pfuetzner[1,2,3,4,5], Axel T Brunger[1,2,3,4,5]*

[1]Department of Molecular and Cellular Physiology, Stanford University, Stanford, United States; [2]Department of Neurology and Neurological Sciences, Stanford University, Stanford, United States; [3]Department of Structural Biology, Stanford University, Stanford, United States; [4]Department of Photon Science, Stanford University, Stanford, United States; [5]Howard Hughes Medical Institute, Stanford University, Stanford, United States; [6]Department of Biochemistry and Molecular Biology, University of Chicago, Chicago, United States

**Abstract** The recycling of SNARE proteins following complex formation and membrane fusion is an essential process in eukaryotic trafficking. A highly conserved AAA+ protein, NSF (*N*-ethylmaleimide sensitive factor) and an adaptor protein, SNAP (soluble NSF attachment protein), disassemble the SNARE complex. We report electron-cryomicroscopy structures of the complex of NSF, αSNAP, and the full-length soluble neuronal SNARE complex (composed of syntaxin-1A, synaptobrevin-2, SNAP-25A) in the presence of ATP under non-hydrolyzing conditions at ~3.9 Å resolution. These structures reveal electrostatic interactions by which two αSNAP molecules interface with a specific surface of the SNARE complex. This interaction positions the SNAREs such that the 15 N-terminal residues of SNAP-25A are loaded into the D1 ring pore of NSF via a spiral pattern of interactions between a conserved tyrosine NSF residue and SNAP-25A backbone atoms. This loading process likely precedes ATP hydrolysis. Subsequent ATP hydrolysis then drives complete disassembly.

DOI: https://doi.org/10.7554/eLife.38888.001

*For correspondence:
brunger@stanford.edu

## Introduction

The archetypal Type II AAA+ protein NSF (*N*-ethylmaleimide-sensitive-factor) plays an essential role in eukaryotic trafficking through its disassembly of different SNARE (Soluble *N*-ethylmaleimide-sensitive factor attachment protein receptor) complexes (*Zhao and Brunger, 2016*). This process has been studied extensively in the context of neurotransmission, where synaptic vesicle fusion with the presynaptic membrane is driven by the formation of the ternary neuronal SNARE complex, an exceptionally stable four-helix bundle composed of syntaxin, synaptobrevin, and SNAP-25 (*Fasshauer et al., 1998*; *Sutton et al., 1998*; *Weber et al., 1998*). After fusion, NSF—together with several molecules of an adaptor protein, SNAP (Soluble N-ethylmaleimide-sensitive factor Attachment Protein)—binds to the *cis* SNARE complex, forming the so-called 20S complex and disassembles it in an ATP-dependent manner (*Hanson et al., 1997*; *Mayer et al., 1996*; *Söllner et al., 1993*; *Zhao et al., 2015*). The free SNARE proteins are then recycled and used for additional rounds of synaptic vesicle formation and fusion (*Mancias and Goldberg, 2007*; *Miller et al., 2011*; *Mossessova et al., 2003*). Moreover, together with Munc18 and Munc13, NSF and SNAPs are part of a quality control system that ensures proper *trans* SNARE complex assembly (*Lai et al., 2017*; *Ma et al., 2013*).

While the kinetics and specificity of NSF-mediated SNARE complex disassembly have been studied extensively (*Cipriano et al., 2013*; *Matveeva et al., 1997*; *Söllner et al., 1993*; *Vivona et al.,*

2013), structural principles underlying SNARE complex recognition, docking, and disassembly have only recently begun to emerge. NSF is composed of an N-terminal domain (N) and two ATPase domains (D1 and D2). Crystal structures of the neuronal SNARE complex, SNAPs, and the NSF N and D1 domains have been determined (*Lenzen et al., 1998*; *May et al., 1999*; *Rice and Brunger, 1999*; *Sutton et al., 1998*; *Yu et al., 1998*), and a cryo-electron microscopy (cryo-EM) study revealed the quaternary structure of both substrate-free full-length NSF in the presence of ATP or ADP at near atomic resolution, as well as the structures of two different 20S complexes at 7 – 8 Å resolution (*Zhao et al., 2015*). Regardless of the nucleotide state or the presence of substrate, six NSF molecules form a two-layer ring structure. While the D2 ring is nearly perfectly six-fold symmetric in all structures determined thus far, the conformation of the D1 ring changes substantially depending on its nucleotide state. These conformational changes are likely related to those sampled during substrate processing; in the ADP-bound state, the D1 ring is in an open flat-washer conformation, while the ATP-bound D1 ring further separates and forms a split-washer conformation (*Zhao et al., 2015*). Structures of the 20S complex in the presence of the non-hydrolysable ATP analogue AMPPNP largely recapitulate the split-washer configuration of the D1 ring (*Zhao et al., 2015*). In the case of the 20S complex formed with the neuronal SNARE complex, the SNAREs are positioned nearly coaxially above the D1 ring, with the N-termini of the SNAREs in the vicinity of the D1 pore. This positioning is accomplished by the NSF N domains and αSNAP—depending on the particular SNARE complex, between two and four αSNAP molecules were present, surrounded by four to six N domains. Together, the N domains, αSNAPs, and SNAREs form a 'spire' on top of the D1 ring.

While these structures of substrate-free NSF and the 20S complex provided first insights into structure of the 20S complex and the conformational changes of NSF associated with the nucleotide state, they ultimately lead to questions about substrate recognition and the conformational cycle associated with NSF-mediated disassembly of the SNARE complex. In terms of substrate recognition, for example, the number of αSNAPs varied based on the particular SNARE proteins used—the structure of the V7-20S complex, composed of NSF, αSNAP, Vamp7, the full cytoplasmic region of syntaxin-1A including the $H_{abc}$ domain, and full-length SNAP-25A contained only two αSNAPs, whereas the structure of the 20S complex composed of NSF, αSNAP, the full cytoplasmic region of synaptobrevin-2, a cytoplasmic fragment of syntaxin-1A without the $H_{abc}$ domain, and the two SNARE motifs of SNAP-25A included four αSNAPs (*Zhao et al., 2015*). Moreover, a 20S complex composed of NSF, αSNAP, the full cytoplasmic region of syntaxin-1A including the $H_{abc}$ domain, and the two SNARE motifs of SNAP-25A also revealed four αSNAPs (*Zhou et al., 2015a*). However, in all of these 20S structures, the local resolution of the reconstructed maps did not permit assignment of the specific SNARE proteins in the density associated with the four-helix bundle. Furthermore, no interaction between the SNAREs and the D1 ring of NSF could be observed, leaving questions related to substrate recognition and processing unanswered.

Here, we present higher-resolution cryo-EM reconstructions of the 20S complex composed of NSF, αSNAP, the full cytoplasmic regions of synaptobrevin-2 and syntaxin-1A, and full-length SNAP-25A in the presence of ATP under non-hydrolyzing conditions. These new reconstructions reveal the interaction between the SNARE complex, the αSNAPs, and NSF in unprecedented detail and permits their contextualization in the disassembly process.

## Results and discussion

### 3D classification identifies multiple conformational states for the 20S complex

The 20S complex was prepared in a manner similar to that described previously (*Zhao et al., 2015*) using hexameric NSF, αSNAP, and nearly full-length soluble neuronal SNARE complex. Two key changes were made, however. First, hexameric NSF was prepared in the presence of ATP instead of the slowly hydrolyzable analogue AMPPNP. Moreover, to prevent hydrolysis, $Mg^{2+}$ was omitted from the buffer, and EDTA was included to remove any trace divalent cations (see Materials and methods). Second, the SNARE complex was prepared using the nearly full-length soluble portions of rat syntaxin-1A (residues 1–256) and synaptobrevin-2 (residues 1–89), with the C-terminal transmembrane segments omitted. Additionally, nearly full-length rat SNAP-25A (residues 1–

204) was used in which the long, unstructured linker spanning the distance between the two SNARE motifs is present.

3D classification produced four classes, two of which (Classes II and IV) contain all components of the 20S complex, while Classes I and III are not well resolved. Refinement of the class that appears to have the best-resolved density for the entire 20S complex (Class IV) without symmetry restraints yielded two reconstructions of 20S complexes (FL-20S-1 and FL-20S-2 at 4.4 Å; *Figure 1*); each reconstruction reveals density for a 20S complex which differs from previous reconstructions in several key ways. As before (*Zhao et al., 2015*), clear density is present for all components of the 20S complex, albeit with interesting variations. Most importantly, a tube of density runs from the SNARE complex to the center of the D1 ring of NSF, suggesting direct engagement of substrate. To improve the detail of the reconstruction, focused refinement (see Materials and methods) was performed, in which all but the NSF D1 and D2 domains was masked; this approach yielded two classes of complexes with near-atomic resolution (FL-20S$_{focus}$-1 and FL-20S$_{focus}$-2, 3.8 Å and 3.9 Å, respectively) (*Figure 1—figure supplement 1*). The focused FL-20S$_{focus}$-1 and FL-20S$_{focus}$-2 classes show well-resolved density for a variety of features, including secondary structure and side chains (*Figure 2*). Models were built into the density starting from structures published previously (*Zhao et al., 2015*); all refinement statistics are summarized (*Table 1*).

## Structures of the ATP-bound 20S supercomplex position substrate for D1 engagement

In both FL-20S classes, the NSF D1 and D2 ATPase domains form a pair of stacked, hexameric rings, with N domains, αSNAP, and soluble neuronal SNARE complex forming a spire-like structure on top (*Figure 3*). Examination of the reconstructed density reveals features consistent with secondary structure and even side chains (*Figure 3A*). This enabled modeling of the majority of the 20S complex (*Figure 3B*). While the D2 ring is largely six-fold symmetric, the D1 ring forms an asymmetric split-washer in which the six D1 domains of protomers A–F undergo a rigid-body transformation in a spiral pattern (average rotation of 57.5° ± 1.0° about the principle axis), with each consecutive counter-clockwise step away from the domain closest to the D2 ring (protomer A). While the D2 domain of protomer F is clearly defined, density corresponding to its D1 domain is poor and not of sufficient quality to permit backbone tracing, implying substantial conformational heterogeneity. A rigid body fit of the equivalent protomer from the ATP-bound NSF structure published previously (*Zhao et al., 2015*) suggests a similar conformation for protomer F in which the domain sits at the top of the D1 helix, furthest from the D2 ring, with its pore loop relatively far from the substrate. Moreover, the small D1 subdomain of protomer F is rotated in a more pronounced fashion than those from the other protomers (*Figure 4*).

Regarding substrate loading, both FL-20S classes reveal a spire in which the N-terminal end of the SNARE complex is positioned over the D1 ring by two αSNAP molecules, each of which is in turn bound by two N domains (*Figure 3A*). Critically, and in contrast to previous studies, the map is of sufficient quality to permit explicit assignment of the four SNARE α-helices and thus the determination of the absolute orientation of the SNARE complex in the FL-20S complex (*Figure 5A*). Each helix is individually resolvable, and a rigid body fit of the structure of the crystal structure of the neuronal SNARE complex (*Sutton et al., 1998*) places it in an orientation which properly accounts for the curvature of the bundle. This assignment is corroborated by the presence of a low-resolution feature that approximately matches the position of syntaxin-1A F216 in the asymmetrical −3 layer of the SNARE complex (*Figure 6*). The other residues of this layer are synaptobrevin-2 M46, SNAP-25A G43 and A164 (*Fasshauer et al., 1998*). These reconstructions do not show any sign of intercalation by αSNAP residues into the SNARE complex as had been previously proposed based on a lower resolution reconstruction (*Zhou et al., 2015a*). The assignment of the SNARE complex components is also supported by sidechain densities of the N-terminal residues of SNAP-25A (*Figure 2C*) and by the presence of diffuse density consistent with the linker that connects the two SNAP-25A SNARE motifs (*Figure 7*). This study for the first time provides structural information about the location of the SNAP-25A linker, albeit at low resolution; although the weak density suggests that it samples multiple conformations, these data show that it is at least partially associated with the SNARE complex.

Knowledge of the absolute orientation of the SNARE complex permits analysis of the interactions between the individual SNAREs and αSNAPs. The primary αSNAP interfaces are formed through

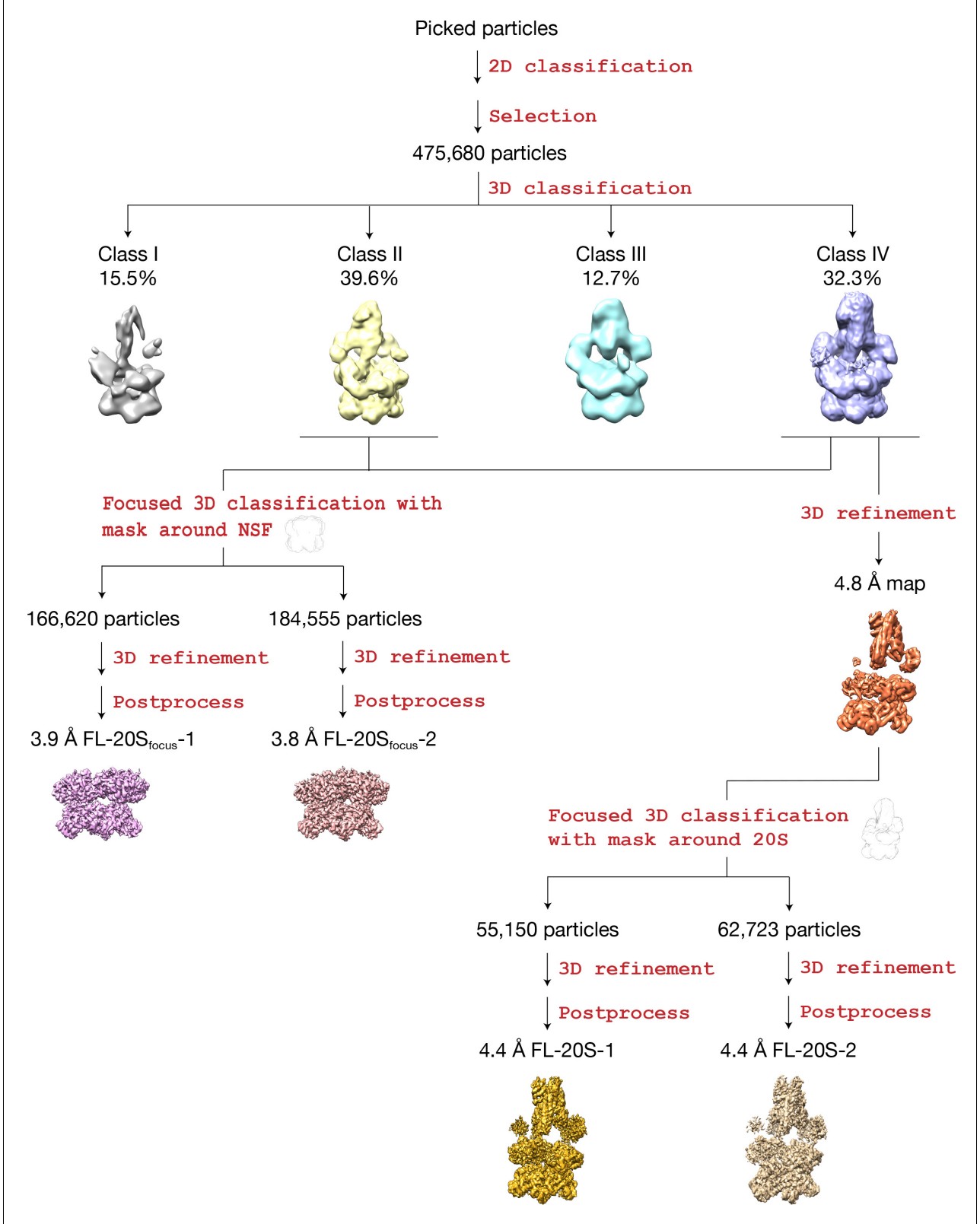

**Figure 1.** Flowchart of data processing. After initial 2D classification, 475,680 particles were selected for 3D classification. Two out of the resulting four classes show significantly better features (Classes II and IV). Class IV shows clear density for N domains, SNARE complex, and αSNAP, while these features are weaker—but still present— in the case of Class II. Two refinement paths were thus taken (Materials and methods). The first path combined

*Figure 1 continued on next page*

*Figure 1 continued*

the two classes and focused on NSF, yielding two maps at 3.9 Å and 3.8 Å with different conformations at the pore loops. The second path focused on Class IV, which yielded two maps at 4.4 Å with different conformations for the SNARE and αSNAP subcomplex.

DOI: https://doi.org/10.7554/eLife.38888.002

The following figure supplements are available for figure 1:

**Figure supplement 1.** Representative data and resolution estimation for single particle analysis.

DOI: https://doi.org/10.7554/eLife.38888.003

**Figure supplement 2.** 3D density maps for (A) FL-20S-1, (B) FL-20S-2, (C) FL-20S$_{focus}$-1, and (D) FL-20S$_{focus}$-2 colored according to local resolution as calculated by ResMap (*Kucukelbir et al., 2014*).

DOI: https://doi.org/10.7554/eLife.38888.004

**Figure supplement 3.** Plots of angular distributions for each of the 20S classes perpendicular to and along the NSF pore axis (A–D).

DOI: https://doi.org/10.7554/eLife.38888.005

electrostatic interactions between two positively charged αSNAP surfaces and a complementary, negatively charged surface formed by the SNARE motifs of synaptobrevin-2, syntaxin-1A, and the second SNARE motif of SNAP-25A (*Figure 5B*). This electrostatic surface distribution is highly conserved for all SNARE complex structures determined to date (*Diao et al., 2015*), suggesting a general principle for SNAP recognition. This SNARE surface is oriented away from the D1 pore; in contrast, the more neutral first SNARE motif of SNAP-25A is largely exposed, facing the D1 ring split.

While both FL-20S classes show clear density for the spire, its orientation varies by a discrete rotation (if viewed from the D1 end, counter-clockwise) about the hexamer axis as well as a slight translation in the hexamer plane away from the split in the D1 ring in each (*Figure 3C*). In the first class, protomers A and B engage one αSNAP, while protomers C and D engage the other (*Figure 3C*, left). In the second class, the N domain is shifted by one protomer counter-clockwise about the

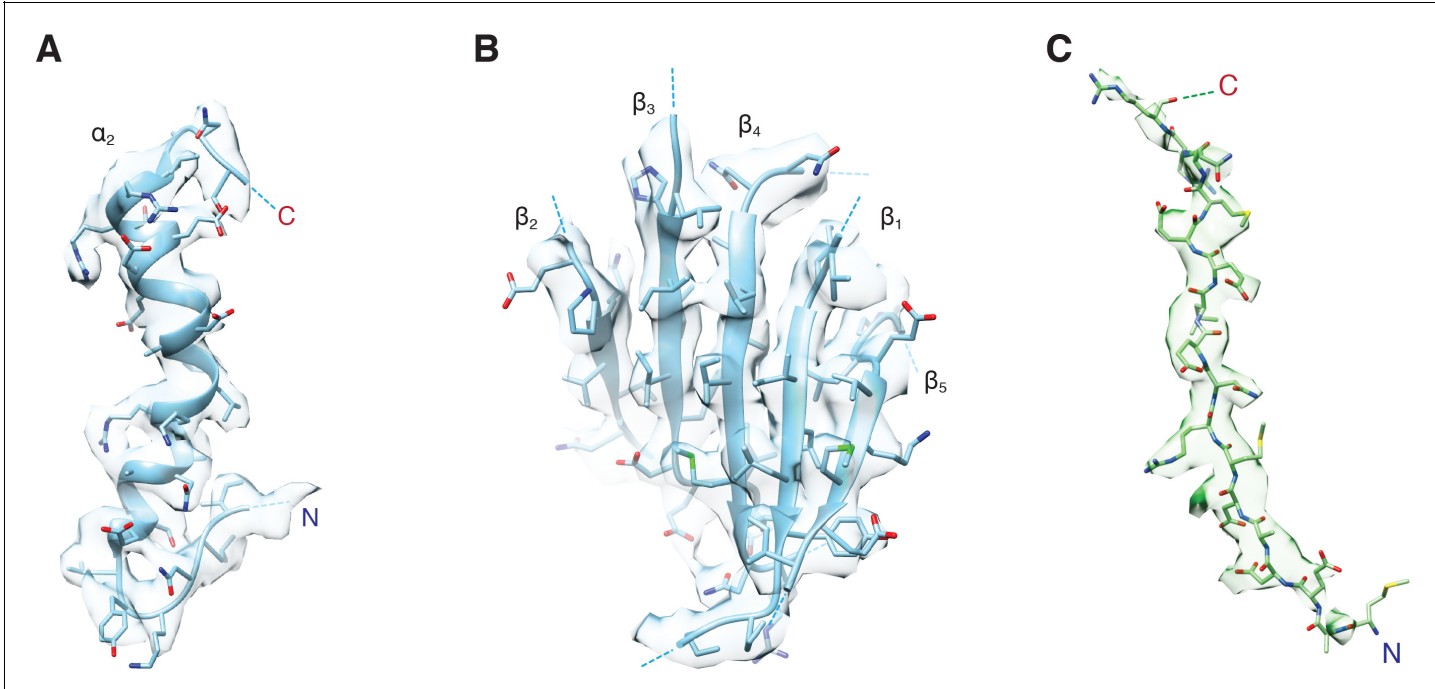

**Figure 2.** Representative density for various features from the FL-NSF$_{focus}$-1 class, contoured at 4.8 σ. (**A**) The α$_2$ helix and pore loop of the protomer B D1 domain, with residues 290 – 320 shown. (**B**) The parallel β sheet from the protomer B D1 domain, composed of β strands 1 – 5. (**C**) The presence of the first 17 residues of SNAP-25A is supported by the FL-20S$_{focus}$-1 density. Density is strongest for residues in the pore of NSF, while the first two N-terminal residues and residues beyond the C-terminus of R17 appear more conformationally heterogeneous.

DOI: https://doi.org/10.7554/eLife.38888.006

**Table 1.** Refinement statistics for each reconstruction and model.

| | FL-20S-1 | FL-20S-2 | FL-20S$_{focus}$-1 | FL-20S$_{focus}$-2 |
|---|---|---|---|---|
| Data acquisition: | | | | |
| Microscope | FEI Titan Krios | FEI Titan Krios | FEI Titan Krios | FEI Titan Krios |
| Detector | Gatan K2 Summit | Gatan K2 Summit | Gatan K2 Summit | Gatan K2 Summit |
| Voltage (keV) | 300 | 300 | 300 | 300 |
| Electron dose (e$^-$ Å$^{-2}$) | 58 | 58 | 58 | 58 |
| Dose rate (e$^-$ sec$^{-1}$ px$^{-1}$) | 10 | 10 | 10 | 10 |
| Pixel size (Å) | 1.31 | 1.31 | 1.31 | 1.31 |
| Defocus range (μm) | 1.5–3.0 | 1.5–3.0 | 1.5–3.0 | 1.5–3.0 |
| Refined particles | 475680 | 475680 | 475680 | 475680 |
| Reconstruction: | | | | |
| Final particles | 166620 | 184555 | 55150 | 62723 |
| Resolution (masked): | | | | |
| FSC, 0.5 (Å) | 6.3 | 6.0 | 4.4 | 4.2 |
| FSC, 0.143 (Å) | 4.4 | 4.4 | 3.9 | 3.8 |
| Resolution (unmasked): | | | | |
| FSC, 0.5 (Å) | 7.5 | 7.3 | 6.1 | 5.7 |
| FSC, 0.143 (Å) | 5.8 | 5.3 | 4.2 | 4.2 |
| Sharpening B-factor (Å$^2$) | –178.83 | –181.52 | –151.16 | –150.50 |
| Model composition: | | | | |
| Total atoms | 73313 | 73052 | 45507 | 45348 |
| Peptide chains | 11 | 11 | 7 | 7 |
| Protein residues | 4662 | 4643 | 2854 | 2843 |
| Refinement: | | | | |
| Unit cell | P1 | P1 | P1 | P1 |
| a, b, c (Å) | 301.30, 301.30, 301.30 | 163.75, 146.72, 227.94 | 145.41, 137.55, 117.9 | 301.30, 301.30, 301.30 |
| α = β = γ (°) | 90 | 90 | 90 | 90 |
| CC$_{mask}$ | 0.77 | 0.77 | 0.80 | 0.78 |
| Resolution (vs. model, masked): | | | | |
| FSC, 0.5 (Å) | 5.9 | 4.8 | 4.0 | 4.0 |
| FSC, 0.143 (Å) | 4.1 | 3.9 | 3.6 | 3.5 |
| Resolution (vs. model, unmasked): | | | | |
| FSC, 0.5 (Å) | 7.0 | 6.8 | 4.3 | 4.3 |
| FSC, 0.143 (Å) | 4.3 | 4.2 | 3.8 | 3.8 |
| RMS Deviations: | | | | |
| Bond lengths (Å) | 0.010 | 0.016 | 0.015 | 0.009 |
| Bond angles (°) | 1.337 | 1.77 | 1.52 | 1.53 |
| Ramachandran statistics: | | | | |
| Favored (%) | 95.82 | 96.94 | 99.65 | 97.51 |
| Allowed (%) | 3.90 | 2.82 | 0.31 | 2.16 |
| Outliers (%) | 0.28 | 0.24 | 0.04 | 0.33 |
| Validation: | | | | |
| Rotamer outliers (%) | 0.10 | 0.36 | 0.16 | 0.33 |

*Table 1 continued on next page*

*Table 1 continued*

|  | FL-20S-1 | FL-20S-2 | FL-20S$_{focus}$-1 | FL-20S$_{focus}$-2 |
| --- | --- | --- | --- | --- |
| All-atom clashscore | 4.75 | 6.63 | 6.69 | 5.27 |
| EMRinger score | 0.51 | 0.52 | 1.91 | 1.39 |
| MolProbity score | 1.53 | 1.55 | 1.37 | 1.38 |

DOI: https://doi.org/10.7554/eLife.38888.007

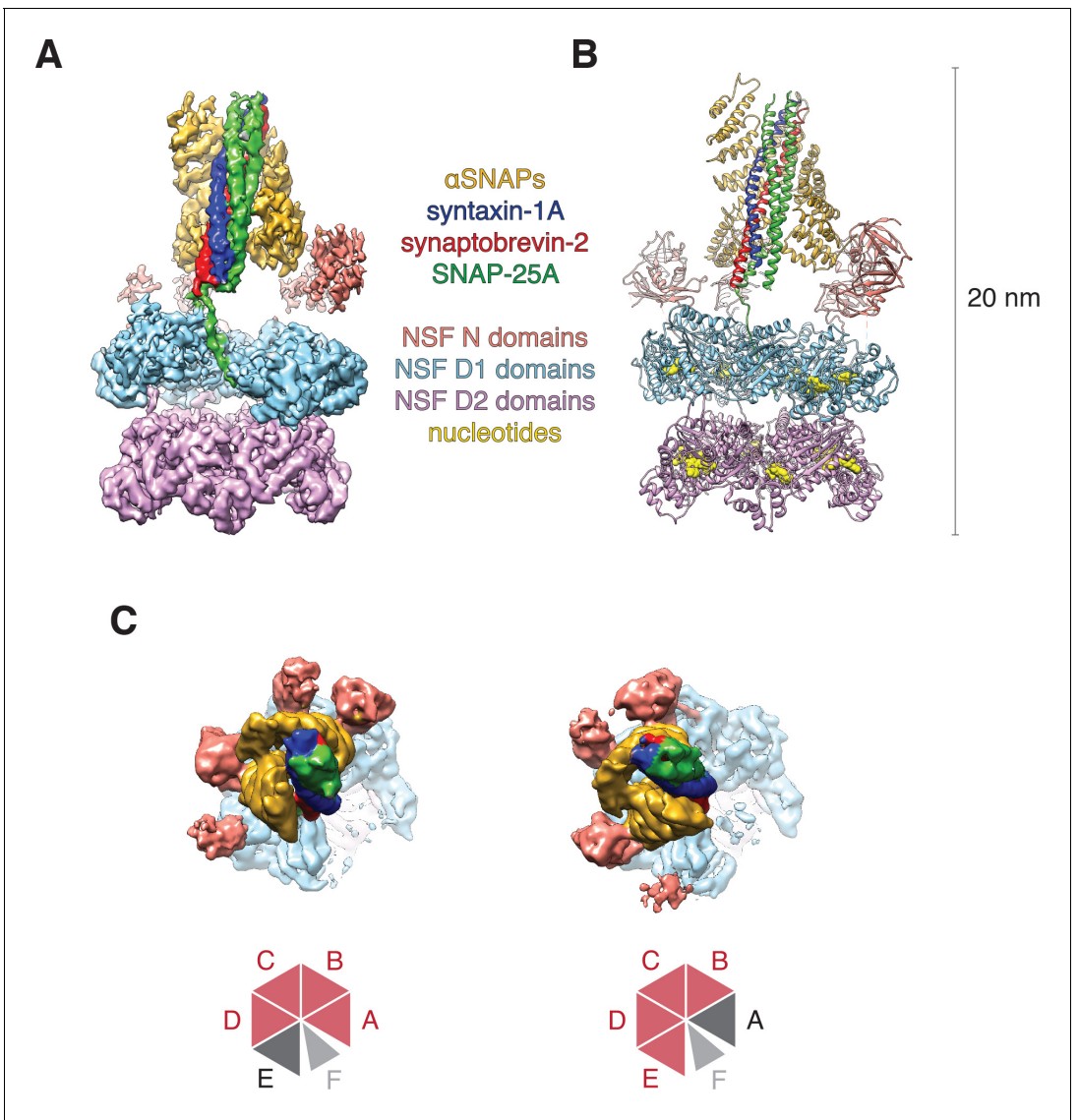

**Figure 3.** Architecture of the 20S complex, composed of NSF (N domains, salmon; D1 domains, cyan; D2 domains, purple), αSNAPs (gold), and the neuronal SNARE complex (syntaxin-1A, red; synaptobrevin-2, blue; SNAP-25A, green). (A) Sharpened FL-20S-1 map contoured at 4.8 σ; N domains for NSF subunits A–D are visible at this threshold. (B) FL-20S-1 composite model, with nucleotides represented by yellow spheres. (C) The pattern of N domain engagement with the αSNAP/SNARE complex varies between the FL-20S-1 and FL-20S-2 classes; in the second class, the pattern of engagement shifts one protomer counter-clockwise about the hexamer axis. The bottom panels show schemas of the configurations. Despite changes in spire architecture, the split in the D1 ring is found between protomers A and F in both classes, with protomer A furthest from the viewer.
DOI: https://doi.org/10.7554/eLife.38888.008

The following figure supplement is available for figure 3:

**Figure supplement 1.** Comparison of NSF N domain engagement with αSNAPs and different SNARE complexes.
DOI: https://doi.org/10.7554/eLife.38888.009

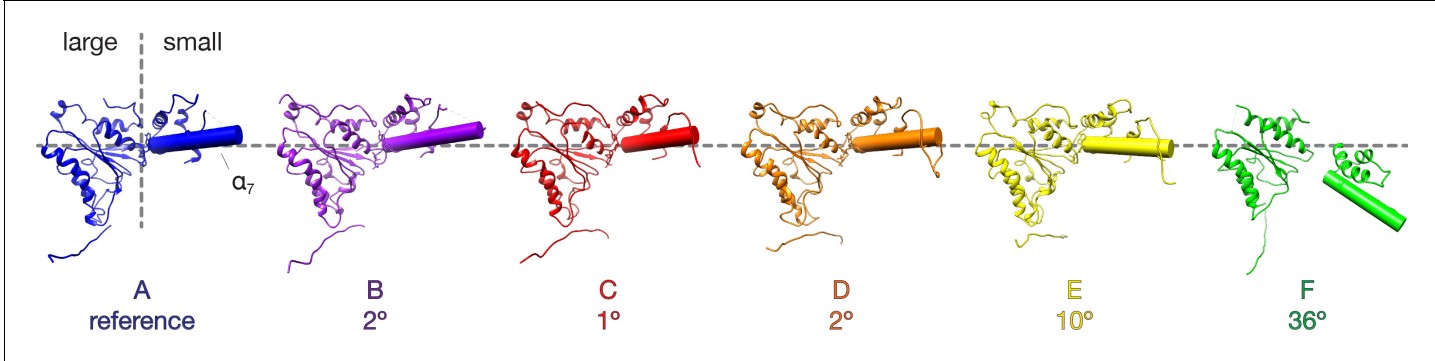

**Figure 4.** Comparison of NSF D1 small subdomain conformations in the FL-20S-1 model. The large D1 subdomains of all protomers were superimposed, and angles were calculated between the small subdomain $\alpha_7$ helical axes of protomer A and protomers B-F. Superposition was performed on the large subdomain due to improved alignment. Protomers E and F show substantial angular deviations in comparison to the domains A-D, consistent with remodeling of the hinge region between the large and small subdomains.
DOI: https://doi.org/10.7554/eLife.38888.010

hexamer axis, with domains B and C engaging the first αSNAP and domains D and E engaging the second (*Figure 3C*, right). This difference results in a 53.2° rotation of the αSNAPs and SNARE complex about the hexamer axis. Despite the variation in spire configuration between these classes, there is little difference in the interaction between SNAP-25A and the D1 pore or in the overall configuration of the D1 and D2 domains of protomers A–E.

Comparison of FL-20S classes with the previous structures of the 20S complex prepared with truncated neuronal SNARE complex (T-20S) (*Zhao et al., 2015*) is also informative. The four T-20S structures published previously differed in the configuration of the spire architecture; six N domains were found in bound or free states surrounding four αSNAPs on the SNARE complex. The structures of FL-20S-1 and FL-20S-2 are most similar to that of T-20S-2 (PDB ID 3J97) and T-20S-3$_a$, respectively, with the N domains of protomers A–D and B–E interacting tightly with two adjacent αSNAPs (*Figure 3—figure supplement 1*). Unlike the T-20S-2 structure, however, the density for the remaining N domains is too weak to model. While this core spire architecture is consistent between the FL-20S and T-20S-2 classes, superposition of the two common αSNAPs shows that the orientation of the SNARE complex axis relative to the D1 ring pore axis varies considerably (*Figure 8*). This difference likely arises from the differing length of the N termini present in the truncated and full-length SNARE complexes, as both the FL-20S and V7-20S spires are pushed off-axis relative to the D1 pore. This may arise through the flexibility of the linkers between the N and D1 domains. The stoichiometry of the spire structure also differs between classes. Previously, four αSNAPs were found associated with all T-20S complex classes (*Zhao et al., 2015*); in the case of the FL-20S reconstruction presented here, only two are present, similar to the structure of the V7-20S complex (*Figure 3C*).

What could account for this difference in αSNAP stoichiometry? Both the V7-20S structure and the FL-20S structures were prepared using full-length SNAP-25A with a long, flexible linker spanning the length of the SNARE bundle and connecting the parallel SNAP-25A SNARE motifs. As mentioned above, density consistent with this linker runs along the solvent-exposed surface of the FL-20S SNARE complex from the C-terminus of the first helix to the N-terminus of the second (*Figure 7*). Density is strongest near the SNAP-25A helical ends and more diffuse towards the center of the SNARE bundle. The presence of this SNAP-25A linker likely interferes with binding by an additional pair of αSNAPs, an observation supported by gel densitometry as well (*Choi et al., 2018*). Furthermore, in all structures of 20S complexes with full-length SNAP-25A determined to date, two rather than four bound αSNAPs are observed. However, for SNARE complexes that do not contain a SNAP-25A linker such as the SNARE complex involved in vacuolar membrane fusion, the number of αSNAP molecules in the initial 20S complex may be higher (*Lobingier et al., 2014*). Moreover, additional αSNAP molecules may be engaged during the hydrolysis cycle (*Shah et al., 2015*).

Finally, and most important, density consistent with the N-terminal residues of SNAP-25A runs from the tip of the SNARE complex to the center of the D1 ring, where it assumes an extended

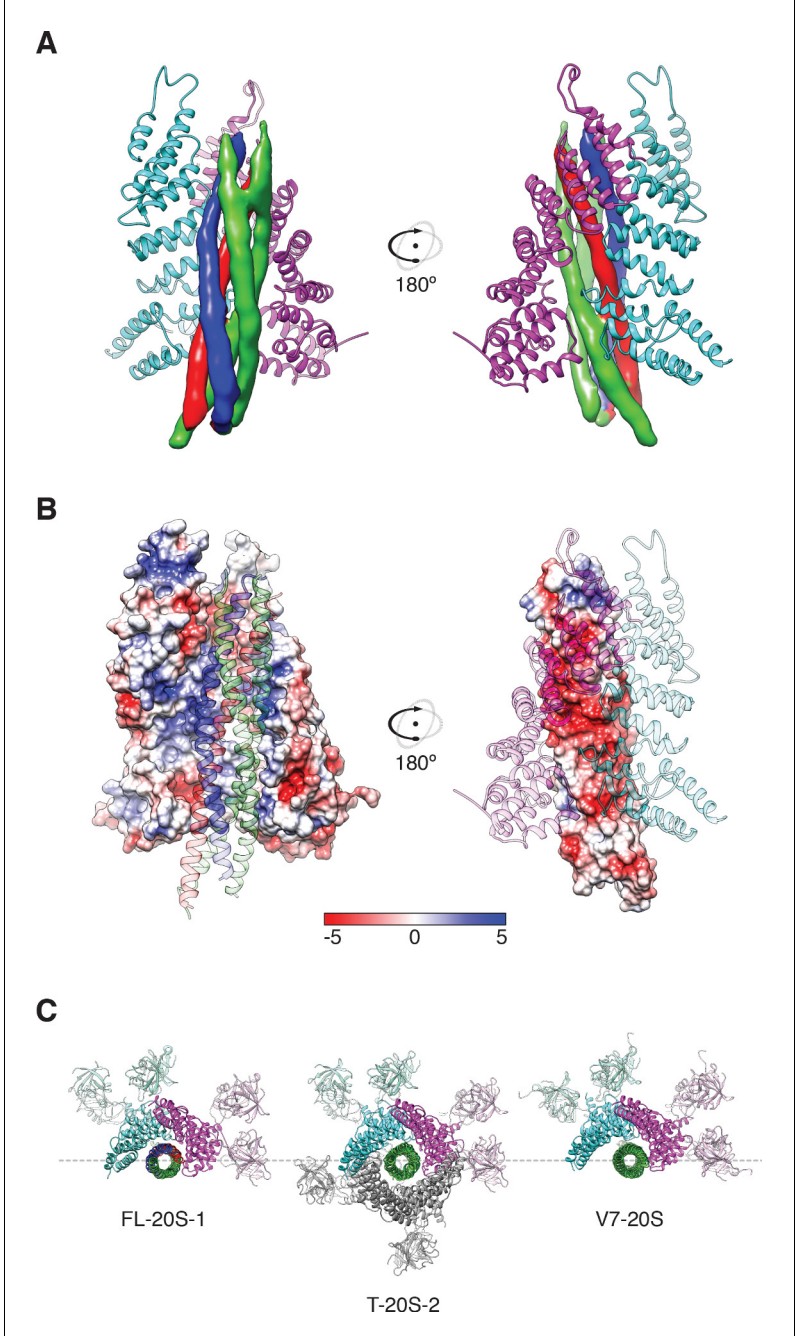

**Figure 5.** Two αSNAP molecules (blue, magenta) bind the full-length neuronal SNARE complex composed of SNAP-25A (green), syntaxin-1A (red), and synaptobrevin-2 (blue) at a well-defined orientation. (**A**) Reconstructed density for the neuronal SNARE complex in the FL-20S-1 map, contoured at 7.1 σ. (**B**) Electrostatic surfaces for the αSNAPs (left) and SNARE complex (right) reveal a complementary pattern of positive and negative charge, respectively. Calculations were performed using APBS assuming 150 mM NaCl, with units in kT. (**C**) Comparison of N domain and αSNAP stoichiometry for the full-length (FL-20S-1) and truncated (T-20S-2) reconstructions as well as for the V7-20S reconstruction; T-20S-2 is the only complex to show four αSNAPs bound to the SNARE complex.
DOI: https://doi.org/10.7554/eLife.38888.011

conformation at a 61.0° angle relative to the plane of the D1 ring (*Figure 3A*). This interaction is preserved in the FL-20S-2 reconstruction as well, with a slight difference in the unstructured region of SNAP-25A running between the structured region of the SNARE complex and the D1 pore entrance. While it has been previously speculated that SNAP-25A is the SNARE complex member upon which

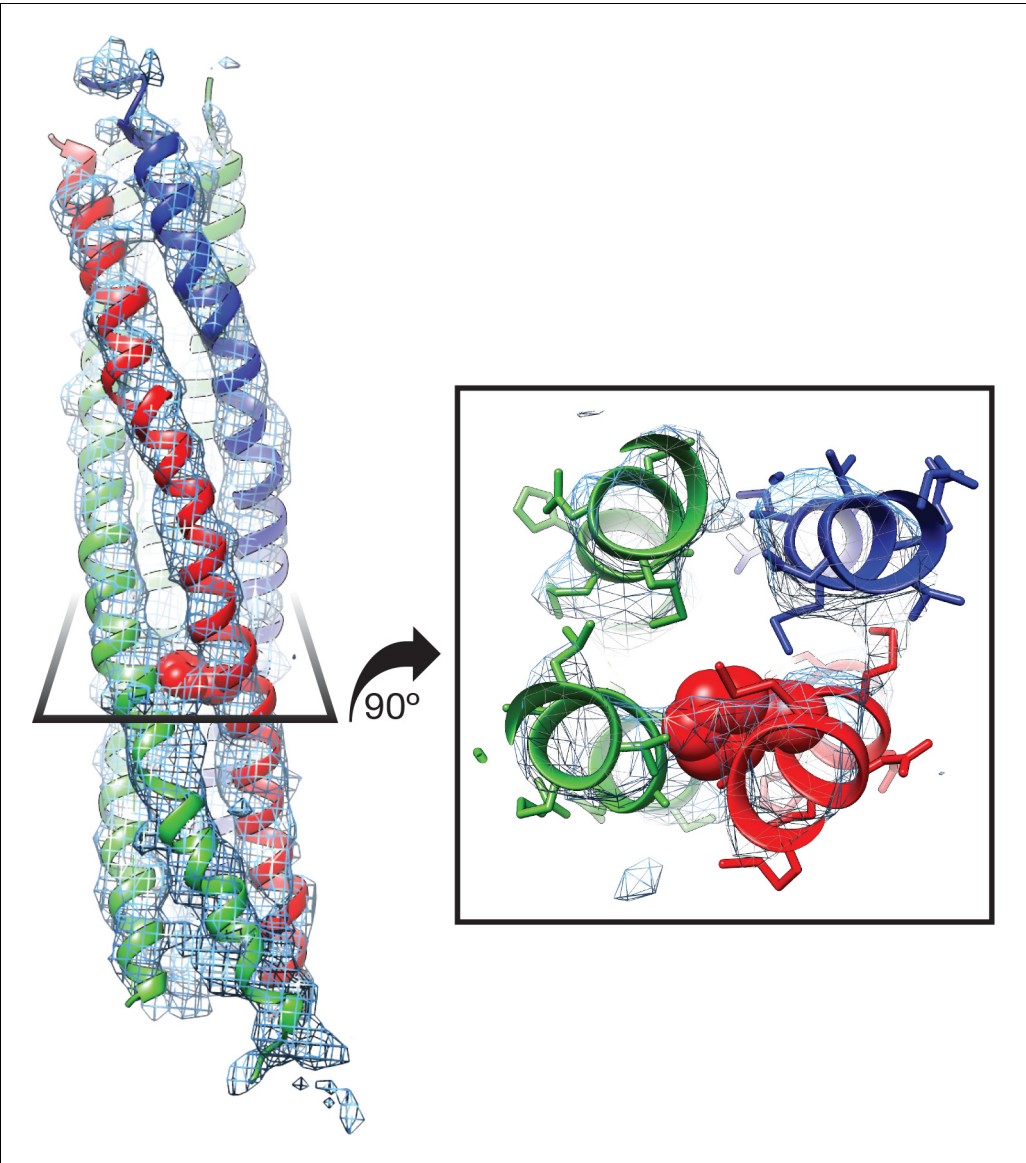

**Figure 6.** Side chain density at −3 layer corresponding to syntaxin-1A F216 supports SNARE complex orientation. SNAP-25A (green), syntaxin-1A (red), and synaptobrevin-2 (blue) are shown with representative density from the FL-20S-1 reconstruction, contoured at 5.5 σ. F216 is shown as spheres, while nearby side chains are depicted as sticks.

DOI: https://doi.org/10.7554/eLife.38888.012

NSF directly acts (*Cipriano et al., 2013*), this is the first direct structural evidence of such an interaction.

## Atomic details of substrate loading

For both focused FL-20S$_{focus}$-1 and FL-20S$_{focus}$-2 reconstructions, the spires and N domains are weak but faintly visible in the unsharpened maps; FL-20S$_{focus}$-1 appears similar in configuration to the FL-20S-2 class, with N domains from protomers B–E engaged with two αSNAPs, while FL-20S$_{focus}$-2 is more similar to the FL-20S-1 class, with N domains from protomers A–D engaged instead. An additional difference between the two FL-20S$_{focus}$ classes is found in the relative position of the D1 ring relative to the D2 ring. Indeed, superposition of the FL-20S$_{focus}$-1 and FL-20S$_{focus}$-2 D2 rings shows that the D1 rings and substrates undergo a relatively subtle rigid body transformation relative to D2,

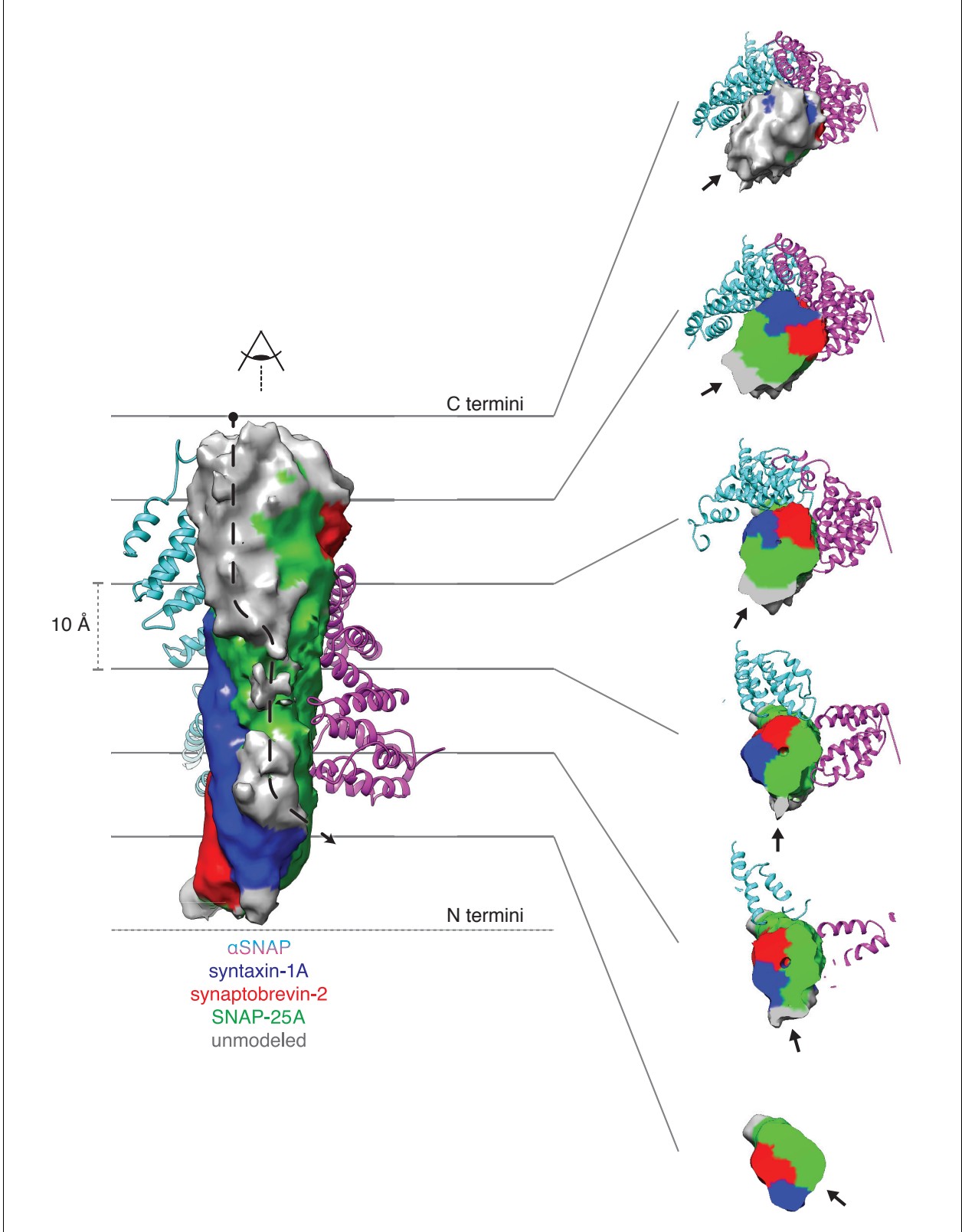

**Figure 7.** Density running along the solvent-exposed face of the SNARE complex is consistent with the presence of the linker that connects the two SNAP-25A SNARE motifs. Surfaces are contoured at 2 σ and colored based on the identity of the nearest atom within 5 Å; green corresponds to SNAP-25A, blue to synaptobrevin-2, and red to syntaxin-1A. Grey surfaces indicate a surface further than 5 Å from any modelled atom. The two αSNAP molecules are depicted in cyan and magenta. The portion of the linker corresponding to the unmodeled density at the top C-terminal end (i.e., the
*Figure 7 continued on next page*

*Figure 7 continued*

membrane proximal region) of the SNARE complex includes four cysteines which are palmitoylated *in vivo* (*Greaves et al., 2009*; *Hess et al., 1992*); this modification promotes association with the plasma membrane during trafficking and enhances its association at the presynaptic membrane. A possible path from N- to C-terminal end of the linker is indicated by a dashed arrow (left). This path is also shown as a series of slices through the complex (right), with an arrow indicating the putative linker density. The relatively weak and diffuse density suggests that the linker is flexible and present in multiple conformations.

DOI: https://doi.org/10.7554/eLife.38888.013

with much of the difference arising from a shift in the location of D1 domain of protomer E and variation in the weak density for the D1 domain of protomer F (*Figure 3C*). Based on this observation and the non-hydrolyzing conditions under which the sample is prepared, the two conformations are likely coupled to the N domain configurations. Given the similarities between these classes, further discussion will focus on the FL-20S$_{focus}$-1 structure.

How does NSF engage substrate, and how does its presence affect the structure of NSF? In the presence of ATP and the absence of substrate, the D1 domains of NSF are arranged in a spiral pattern, with the D1 domain of protomer A closest to the surface of its D2 domain and the D1 domain of protomer F at the top, furthest from the corresponding D2 domain (*Figure 9A*). A pore is formed in the center of the hexamer, with a highly conserved loop (i.e., the pore loop, 294–296, YVG) at the center. The SNAP-25A N-terminal residues are present within this pore in an extended conformation reminiscent of a β-strand with side chain density and close packing with NSF, enabling identification of the register in the density maps (*Figure 2C*). The presence of the SNAP-25A substrate leads to remodeling of the pore loops, with a majority of the conformational change arising through rearrangement of a highly conserved amino acid residue, tyrosine 294. In the absence of substrate, each tyrosine adopts a different conformation as assessed by comparison with the cryo-EM structure of NSF alone (*Zhao et al., 2015*), but in the presence of substrate, each tyrosine flips up and away from the pore axis, forming a spiral pattern (*Figure 9A*). In the case of protomers B–D, each tyrosine Cδ$_1$ atom intercalates nonspecifically into every other space between amino acid side chains of the substrate and is engaged in a stereotyped hydrogen bonding interaction with the nearest substrate carbonyl (*Figure 9C*). When the D1 large subdomains are superimposed, the pore loop conformations of protomers B–E are nearly identical, while protomer A is relatively closer to the D2 ring (*Figure 9B*).

## Mechanistic implications of the FL-20S structure

We tested the functional role of the conserved tyrosine residue in the D1 pore. As is the case in other AAA+ proteins (*Chang et al., 2017*), disruption of this interaction decreases or eliminates disassembly activity. For NSF, a somewhat conservative pore loop mutation (Y294L) reduces disassembly of SNARE complex by 88% per unit time, while a more pronounced mutation (Y294A) reduces disassembly even further, by 94%—both significant changes relative to wild type ($p<0.01$) and, in the case of Y294A, nearly statistically indistinguishable from uninitiated reaction with wild type NSF ($p=0.06$; *Figure 10*). These mutations do not affect the intrinsic rate of ATP hydrolysis as measured by an ATPase activity assay (see Materials and methods; wild type, $28 \pm 4$ ATP min$^{-1}$; Y294L, $26 \pm 2$ ATP min$^{-1}$; Y294A, $31 \pm 5$ ATP min$^{-1}$; $p>0.05$ for both), nor do they disrupt the formation of stable hexamer, suggesting that—as in other AAA+ proteins—the pore loops and the Y294 side chains in particular are essential to the mechanical action of NSF.

It is important to note that these observations are likely not specific to the substrate studied here, as NSF disassembles complexes composed of a variety of SNAREs with different N-termini, both native and engineered (*Cipriano et al., 2013*; *Vivona et al., 2013*). Although densities are visible that match the specific sequence of the SNAP-25A N-terminal residues used for the EM reconstructions of the FL-20S complexes (*Figure 2C*), the Y294 engagement pattern perhaps only depends on the presence of a simple β-strand. This suggests that, following αSNAP/SNARE binding, substrate engagement proceeds generically with intercalation between the N-terminal residues of SNAP-25A. Furthermore, the substrate is not threaded arbitrarily far, but instead assumes a unique register. Although binding to syntaxin-1A or synaptobrevin-2 seems unlikely given the large number of additional residues at either N-terminus, engagement of another SNARE component cannot be ruled out.

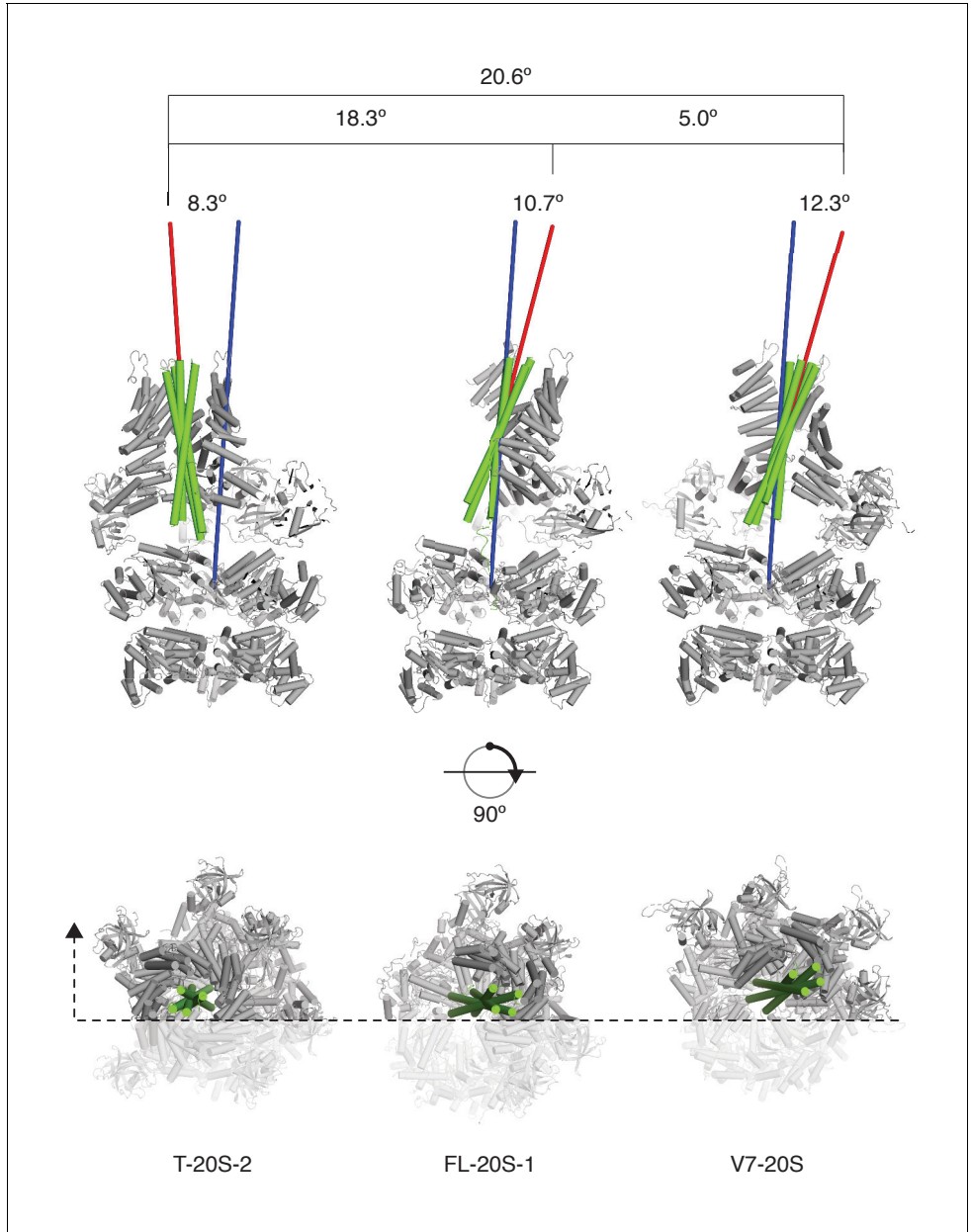

**Figure 8.** SNARE complex orientation varies across constructs. The D1 domains of T-20S-2, FL-20S-1, and V7-20S structures were used to superimpose the rest of each model. SNARE complexes (green) are shown in green relative to the rest of each 20S complex (grey), with the viewing plane slicing through the center of the complex as indicated by a dashed line. Principle axes for each independent SNARE complex (red) and for all D1 domains (blue) are drawn, projecting out from the center of mass of each. The distances between the D1 center of mass and the SNARE complex center of mass, distances between the different SNARE complex centers of mass, angles between SNARE complex principle vectors and the D1 principle vector, and the angles between different SNARE complex principle vectors are shown.

DOI: https://doi.org/10.7554/eLife.38888.014

To assess the specificity of the system for the N-terminus of SNAP-25A, the disassembly assay was repeated using a mutant in which the first 16 acids were removed (SNARE$_{\Delta 16}$ complex). Surprisingly, this truncation slightly increased the rate of SNARE complex disassembly in vitro by 15% as determined by fluorescence dequenching (p<0.01) (*Figure 10*) and SDS-PAGE (*Figure 10—figure supplement 1*). Given the size of the syntaxin-1A and synaptobrevin-2 N-terminal domains, SNAP-25A remains the likely candidate for engagement despite the truncated N-terminus. This result implies primary sequence promiscuity in disassembly and is consistent with previous observations in

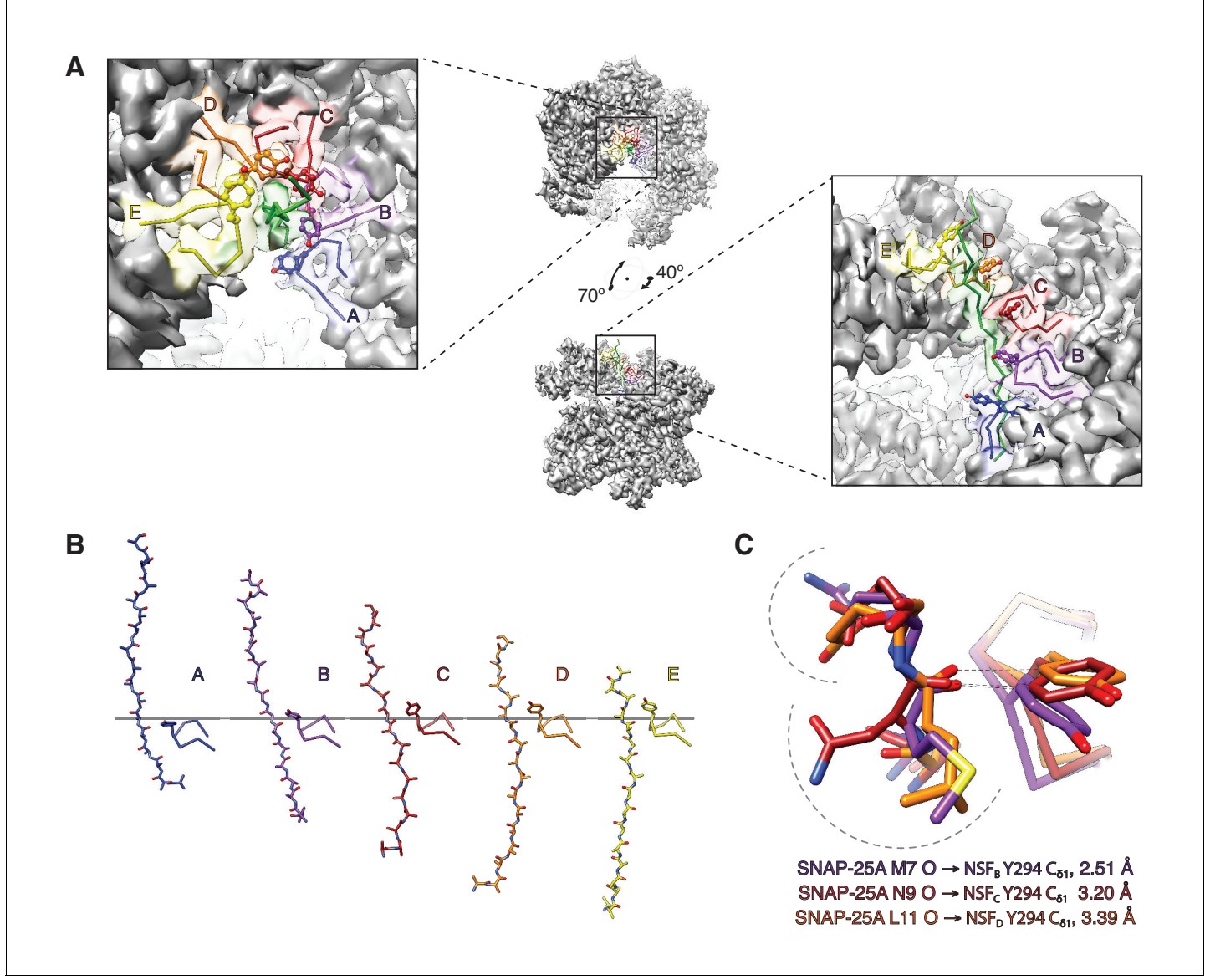

**Figure 9.** The NSF D1 pore loops guide the SNAP-25A N-terminus through the pore. (A) The FL-20S$_{focus}$-1 map, contoured at 4.8 σ, with the SNAP-25A N-terminus and NSF D1 pore loops shown in cartoon format, color coded by protomer. The Y294 side chains are shown as well. The NSF D1 pore loops form a spiral pattern of interactions with the substrate, wherein Y294 interacts with every other substrate residue. The final pore loop from protomer F disengaged. (B) Sequential engagement of SNAP-25A by NSF D1 pore loops on protomers A–E. Full side chains are omitted from the SNAP-25A for clarity. The D1 domains of protomers A–E were superimposed, revealing conformational changes in the pore loops themselves. Protomer A and E pore loops are less directly engaged with substrate, with the protomer A relatively closer to the SNAP-25A N-terminus than expected by symmetry. (C) Overlay of the D1 domain pore loops of protomers B–D reveal a set of stereotyped interactions between each Y294 C$_{δ1}$ and the carbonyl of SNAP-25A residue 7, 9, or 11, respectively. Side chains are positioned away from the intercalation point (dashed lines).
DOI: https://doi.org/10.7554/eLife.38888.015

The following figure supplement is available for figure 9:

**Figure supplement 1.** The secondary NSF D1 domain pore loops (residues 338–345) guide the N-terminus of SNAP-25A through the pore.
DOI: https://doi.org/10.7554/eLife.38888.016

which NSF disassembles different SNARE complexes at similar rates (*Cipriano et al., 2013*; *Vivona et al., 2013*). More generally, this observed promiscuity supports the idea that NSF must disassemble a variety of SNARE complexes in different trafficking contexts in vivo.

The disassembly process likely proceeds without active engagement by a secondary pore loop in the D1 domain. In previous EM structures of NSF, this flexible secondary pore loop (residues 338–

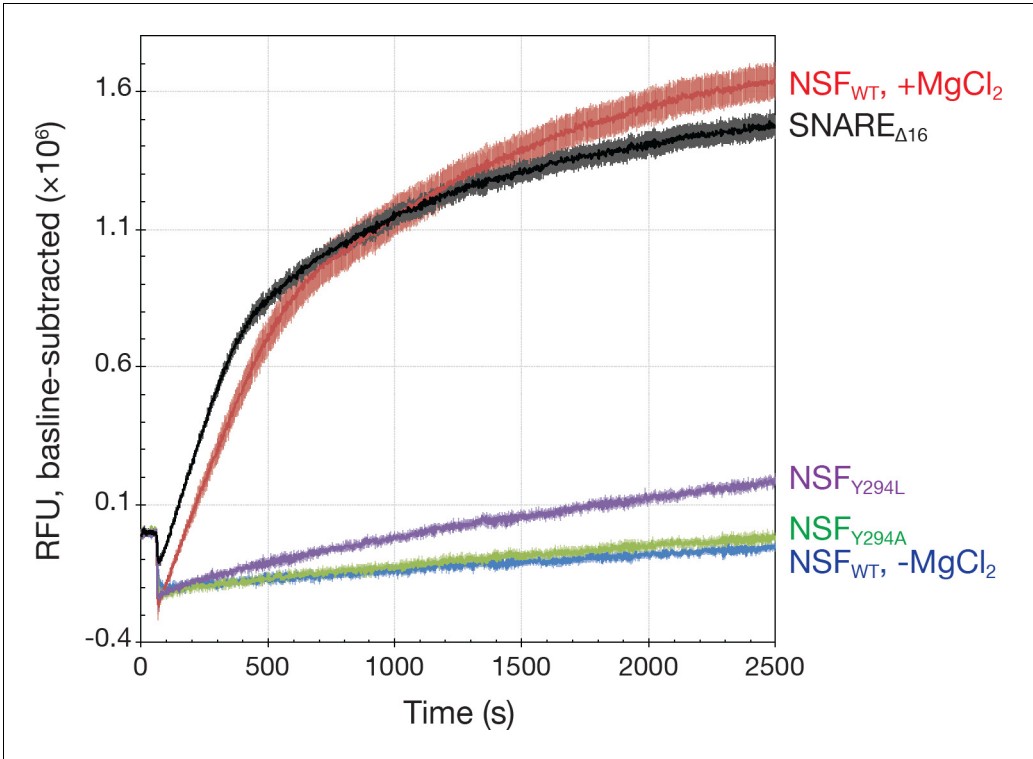

**Figure 10.** Kinetics traces measured using a fluorescence dequenching assay reveal the effects of pore loop mutations and the truncation of the 16 SNAP-25A N-terminal residues (SNARE$_{\Delta16}$) on disassembly activity. Wild-type αSNAP was used throughout. For pore loop mutations and wild type controls, full-length soluble SNARE complex labeled with Oregon Green 488 was used as substrate. For testing disassembly of the SNARE$_{\Delta16}$ complex, wild type NSF was used. All reactions were triggered with the addition of MgCl$_2$. Each curve is the average of five or six replicates; error bars represent standard error about the mean. Mutation of Y294 reduces disassembly rate, while truncation of the SNAP-25A N-terminal residues increases it slightly.

DOI: https://doi.org/10.7554/eLife.38888.017

The following figure supplement is available for figure 10:

**Figure supplement 1.** Gel-based assay for disassembly of full-length neuronal SNARE complex and the truncated neuronal SNARE complex composed of full-length syntaxin-1A, synaptobrevin-2, and SNAP-25A$_{\Delta16}$.

DOI: https://doi.org/10.7554/eLife.38888.018

---

345, GSMAGSTG) in the large subdomain was unresolved (*Zhao et al., 2015*). Here, in the FL-20S$_{focus}$ classes, this loop is present in some subunits and packed to varying degrees against the substrate (*Figure 9—figure supplement 1*). Unlike the primary pore loop, this interaction appears degenerate; different loop residues contact substrate in various locations, suggesting a role in guiding the incoming substrate and/or preventing non-productive interactions with the interior surface of the D1 ring. This is in contrast to other AAA+ proteins, such as the homolog YME1, in which an additional tyrosine on the secondary pore loop also engages with substrate (*Puchades et al., 2017*).

The nucleotide occupancies of the D1 ring in the FL-20S$_{focus}$-1 and FL-20S$_{focus}$-2 maps are consistent with a structure poised to disassemble substrate, in which protomers B–D are ATP-bound (*Figure 11A*, *Figure 11—figure supplement 1*), and key amino acids are in place to catalyze hydrolysis (*Wendler et al., 2012*) (despite the absence of Mg$^{2+}$ under the non-hydrolyzing conditions of the sample preparation [*Figure 11C–E*]). Nucleotide densities associated with the D1 domains of protomers A and E are ambiguous (*Figure 11B,F*). If these densities correspond to ADP, the compound would have been present as a contaminant in the ATP used given the non-hydrolyzing condition of the sample preparation. However, another explanation for the ambiguous nucleotide density may relate to the lower resolution of the FL-20S$_{focus}$-1 and FL-20S$_{focus}$-2 maps for the A D1 domain nucleotide binding pocket and for the entire protomer E D1 domain (*Figure 1—figure supplement 2*). Furthermore, despite possible differences in nucleotide states for protomers A and E, few

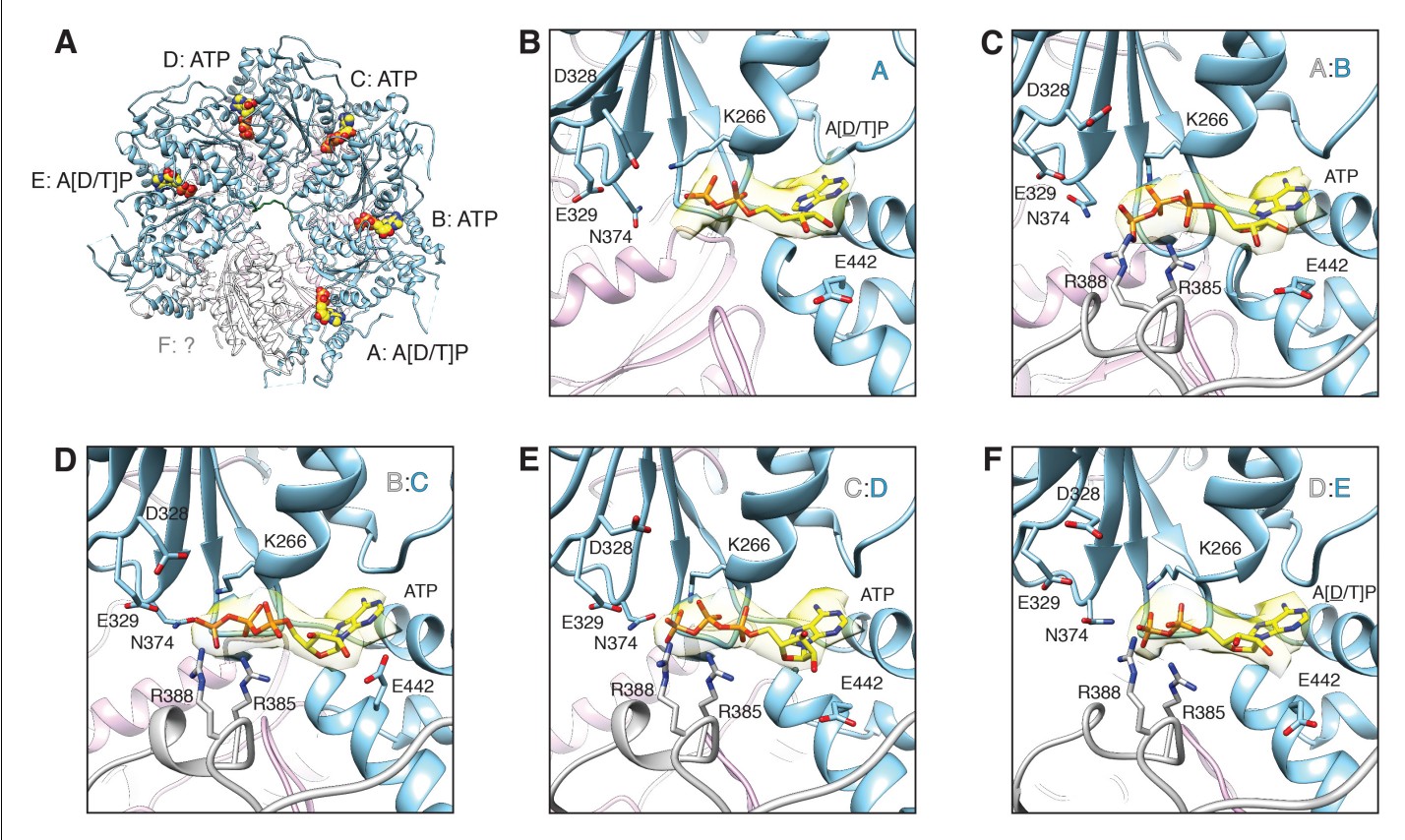

**Figure 11.** FL-20S$_{focus}$-1 D1 domain nucleotide state varies as a function of protomer identity. (**A**) Overview of the D1 ring nucleotide state. D1 domains of protomers B–D (blue) bind ATP while the D1 domains of protomers A and E show ambiguous density. The D1 domain of protomer F from the ATP-bound structure of substrate-free NSF (white) was placed by superposition of D1 rings. Substrate (green), and the D2 ring (purple) are shown for reference. (**B–F**) Nucleotide (yellow) and the nucleotide binding site are shown for each D1 domain. The preceding D1 protomer is also shown (grey). Nucleotide density is contoured at 4.8 σ. In each case, the Walker A motif (p-loop; residues 260–267, including K266) largely coordinates binding. A pair of arginine side chains (R385, R388) from the preceding protomer serve as the arginine fingers and contribute to nucleotide coordination as well. Several essential residues from sensor 1 (N374) and 2 (E442) motifs are also shown. The acidic residues of the Walker B motif (residues D328 and E329 from residues 324–329) are disengaged from the active site in the absence of a magnesium ion.

DOI: https://doi.org/10.7554/eLife.38888.019

The following figure supplement is available for figure 11:

**Figure supplement 1.** Non-segmented density and corresponding model of the D1 nucleotide binding pocket (blue) and bound ATP (yellow) for protomer C from the FL-20S$_{focus}$-1 class reconstruction.

DOI: https://doi.org/10.7554/eLife.38888.020

structural differences are apparent relative to D1 protomers B–D. Protomer E does show some limited conformational changes to the intersubunit signaling (ISS; 359–361, DGV) motif. Finally, weak density for protomer F is consistent with a nucleotide-free state as observed in the ATP-bound, substrate-free cryo-EM structure of NSF (*Zhao et al., 2015*), but nucleotide may nevertheless be present. The D1 ring is quite similar to that this structure (D1 all-atom RMSD = 1.2 Å$^2$; PDB 3J94). While the substrate-bound structure is slightly more expanded about the D1 axis, conformations of essential ATPase elements—for example, the arginine fingers, Walker A/B motifs—are similar overall.

## Relationship to other AAA+ proteins and models for SNARE substrate loading and subsequent processing

Structures of a number of related AAA+ proteins engaged with protein substrate are available— VAT (*Ripstein et al., 2017*), HSP104 (*Gates et al., 2017*), ClpB (*Deville et al., 2017*), Vps4 (*Han et al., 2017*), YME1 (*Puchades et al., 2017*), and TRIP13 (*Alfieri et al., 2018*). As in the case of

NSF, each structure shows substrate engaged by ATPase pore loops at the center of a spiral-shaped hexamer. These pore loops are distinguished by the presence of an apical aromatic amino acid—tyrosine or tryptophan—which intercalates into every other space between side chains of the substrate, gripping it and helping to maintain it in an extended β-strand conformation within the pore. The recurrent observation of this structural pattern has led to the proposal of a processive, 'hand-over-hand' model for translocation in AAA+ in which substrate is presumably pulled further into the pore in a coordinated manner with hydrolysis of ATP. This hydrolysis then leads to changes in the angle between large and small subdomains, effectively moving the seam of the hexamer around the ATPase ring.

Based on our structures, we propose that this generalized hand-over-hand model is consistent with initial substrate loading in the absence of hydrolysis. Indeed, in the absence of $Mg^{2+}$, NSF-mediated disassembly of SNARE complexes was not observed by multiple independent assays (*Figure 10*). This observation suggests that catalysis is not required for initial engagement of substrate. More generally, our proposal of hydrolysis-free substrate loading is also corroborated by the other structures of AAA+ proteins noted above, where only very limited or no nucleotide hydrolysis could have occurred. In the case of ClpB (*Deville et al., 2017*), Hsp104 (*Gates et al., 2017*), and TRIP13 (*Alfieri et al., 2018*), the slowly hydrolyzable analog ATPγS was used and was observed throughout each AAA+ ring, with TRIP13 as the exception lacking nucleotide density for protomer F. Catalytically inactive mutants were used for the structures of TRIP13 and ClpB. The structure of Vps4 (*Han et al., 2017*) was determined in the presence of ADP•BeF$_X$; ADP was observed in protomers A and B, ADP•BeF$_X$ was found in protomers C–E, and protomer F was apparently nucleotide free. Finally, in the case of YME1, the structure of a catalytically inactive mutant was determined in the presence of ATP, although protomer F was observed in an apo-like state and protomer A appeared bound to ADP. In summary, no related, catalytically active AAA+ protein structure has thus far been determined in the presence of substrate under hydrolyzing conditions. As such, it is important to distinguish between a docked, pre-disassembly configuration of a AAA+-substrate complex and an as-of-yet unobserved series of transition states associated with translocation and disassembly.

So, how could such a loaded state form in the absence of hydrolysis? Consider the case of SNAP-25A engagement by NSF. Given a newly formed 20S complex, some initial threading or even substrate unfolding might occur in the absence of hydrolysis. This notion is supported by the observation that the binding of the SNAP-25A N-terminal residues likely involves a conformational change because the SNAP-25A N-terminal residues are otherwise α-helical starting at residue seven as assessed by a crystal structure lacking crystal packing contacts in this region (PDB ID 5W5D) (*Zhou et al., 2015b*); these same residues are extended in the FL-20S complexes (*Figure 3*). If this unfolding occurs at the same time as interaction with NSF, it might be driven by the formation of favorable interactions between the pore loops and the substrate. Alternatively, some degree of 'side-loading' could take place. Given the conformational heterogeneity observed for the large subdomain of the final, highest protomer F of the D1 ring—which does not engage the substrate—and the solvent-oriented configuration of the lowest (first) protomer A, it is plausible that breathing motions of the ring might allow the substrate to pass into the pore at the seam.

It is likely that loading of the SNARE substrate by the D1 pore of NSF would be an important factor to stimulate ATPase activity (*Cipriano et al., 2013*). Upon subsequent ATP hydrolysis, the SNARE substrate could be further threaded into the D1 pore, although complete threading is unlikely as the D2 domain pore is occluded in both ATP- and ADP-bound states (*Zhao et al., 2015*) and palmitoylation anchors SNAP-25A to the membrane following its first SNARE motif (*Greaves et al., 2009*; *Hess et al., 1992*). As such, ATP hydrolysis and large-scale rearrangement of the D1 domains would also impose a force on the N domains of NSF and the attached αSNAP molecules. The combined effect of the N domains and αSNAPs in conjunction with the D1 pore interactions could impose additional shearing force or drive an unwinding process as well (*Zhao et al., 2015*). More work will be required to test these hypotheses.

## Conclusions

Together, the structures of the 20S complex presented here reveal key pre-disassembly states of the 20S complex, in which several degenerate configurations of αSNAPs and NSF N domains position the SNARE complex for engagement by the highly conserved tyrosine residues on the NSF D1 pore loops under non-hydrolyzing conditions. Despite the variable participation of different N domains

about the NSF hexamer, the 2:1 αSNAP:SNARE complex interface is preserved, revealing what is likely a conserved, core electrostatic interaction required for the general formation of the 20S complex. More important, these structures provide evidence for loading of the full-length neuronal SNARE complex via the N-terminal residues of SNAP-25A. Complete disassembly could then proceed through further translocation of the N-terminus of SNAP-25A. Such translocation would exert a pulling force on the remaining membrane-anchored SNAP-25A, likely destabilizing interactions with syntaxin-1A and synaptobrevin-2. Finally, the conditions under which the 20S complex was prepared in addition to the observed nucleotide state of these structures is consistent with—but not direct evidence of—a two-step disassembly mechanism in which the SNARE protein is loaded passively prior to hydrolysis and disassembly. Future studies of conformational intermediates will be essential in testing these models and for exploring possible roles for αSNAP in disassembly.

## Materials and methods

### Protein expression and purification

Wild type and mutant (Y294A, Y294L) NSF from the Chinese hamster *C. griseus* was expressed in BL21(DE3)-RIL *E. coli* and purified as described previously (*Zhao et al., 2015*), with final reassembly of hexameric NSF in a buffer composed of 50 mM Tris pH 8.0, 150 mM NaCl, 1 mM EDTA, 1 mM ATP, and 1 mM TCEP. Rat αSNAP was expressed in *E. coli* and purified as described previously (*Cipriano et al., 2013*). Soluble rat neuronal SNARE complex for the cryo-EM studies (wild-type SNAP-25A 1–204, syntaxin-1A 1–256, and 6×His-synaptobrevin-2 1–89) and for the disassembly assay (wild-type SNAP-25A 1–206 or SNAP-25$_{\Delta16}$ 17–206, syntaxin-1A 1–265, and 6×His-synaptobrevin-2 1–96) were co-expressed in C41 *E. coli* and purified as described previously (*Cipriano et al., 2013*). The 20S supercomplex was formed by mixing ATP-bound NSF, αSNAP, and SNARE complex to a ratio of 1:10:2 under non-hydrolyzing conditions in 50 mM Tris pH 8.0, 150 mM NaCl, 1 mM EDTA, 1 mM ATP, and 1 mM TCEP, and then purified by size-exclusion chromatography using a Superdex 200 10/300 GL column, where the first peak to elute was collected and concentrated to around 15 mg mL$^{-1}$ for cryo-EM analysis.

### Grid preparation and sample vitrification

Quantifoil R1.2/1.3 200 mesh copper grids (Quantifoil Micro Tools GmbH, Germany) were treated with chloroform for 1 hr and dried overnight. Grids were not glow discharged. Prior to freezing, Nonidet P-40 was added to fresh 20S supercomplex to a final concentration of 0.05% v/v as described previously (*Zhao et al., 2015*) to prevent aggregation and enrich for side-views. 2.5 μL volumes were then transferred to grids, blotted for 3–4 s, and plunge frozen in liquid ethane using an FEI Vitrobot (ThermoFisher Scientific, USA).

### Cryo-EM data collection

Cryo-EM data were collected at the Janelia Research Campus cryo-EM facility. Grids were transferred to an FEI Titan Krios (ThermoFisher Scientific, USA) operated at 300 kV. Images were recorded on a Gatan K2 Summit direct electron detector operated in super-resolution counting mode following an established dose fractionation data acquisition protocol (*Li et al., 2013*). The dose rate on the detector was set to be ~10 electrons per pixel per second. The total exposure time was 10 s, leading to a total accumulated dose of 58 e$^-$/Å$^2$ on the specimen. Dose-fractionated images (40 frames) were recorded using SerialEM (*Mastronarde, 2003*). Defocus values ranged from −1.5 to −3.0 μm. Detailed information is summarized in *Table 1*.

### Image processing

Super-resolution counting images were 2 × 2 binned and motion corrected using MotionCorr2 (*Li et al., 2013*). Defocus values and the contrast transfer function were determined for each micrograph using Gctf (*Zhang, 2016*). Subsequent processing was performed using RELION (*Scheres, 2012*). A map of the 20S supercomplex from the previous study (*Zhao et al., 2015*) was used as the initial model. No symmetry was assumed throughout the entire process. Detailed information is summarized in *Table 1*. The classification and refinement workflow are summarized in *Figure 1*. Briefly, after initial 2D classification and cleaning, a total of 475,680 particles were subjected

to 3D classification. Different numbers of classes were tried (3–5), but the results were similar—two classes with well-defined density of the entire 20S complex were identified, one with a less well-resolved spire (*Figure 1*; Class II) and the other with well-resolved spire (*Figure 1*; Class IV). In order to better resolve the ATPase rings of NSF, Classes II and IV were combined and further classified using a mask around the D1 and D2 domains. Different numbers of classes were tried, but two different conformations always emerged. These were refined, yielding two maps with resolutions of 3.9 Å and 3.8 Å, called FL-20S$_{focus}$-1 and FL-20S$_{focus}$-2 (*Figure 1*). Class IV alone was refined to a map of 4.8 Å with well-resolved spire density. With further classification using a tighter mask around the 20S complex, Class IV could be divided into two classes with different conformations (FL-20S-1 and FL-20S-2) and refined to two maps of 4.4 Å resolution, respectively. Other combinations were also tried, but the results always supported two conformations with similar or slightly worse resolution. Focused refinement of the spire was attempted but proved unsuccessful.

## NSF model building and refinement

All model building and manual refinement was performed using Coot (*Emsley et al., 2010*), and automated refinement was carried out using *phenix.real_space_refine* (*Adams et al., 2010*; *Afonine et al., 2018*). No symmetry restraints were imposed at any stage of refinement. Initial models of FL-20S$_{focus}$-1 and FL-20S$_{focus}$-2 were prepared by performing rigid body fits of D1 and D2 domains from the structure of substrate-free, ATP-bound NSF published previously (PDB ID 3J94) (*Zhao et al., 2015*) into sharpened maps. A round of manual model-building was then performed to correct large-scale differences in structure and to extend the model into newly resolvable regions. Automated real space refinement (global minimization, local grid search, ADP refinement) was then performed without secondary structure or Ramachandran restraints. Next, nucleotides and the SNAP-25A 17 N-terminal residues as well as previously unresolved regions of NSF (*Zhao et al., 2015*) were built de novo using Coot. Further automated refinement was performed with secondary structure restraints added.

At this stage, Ramachandran statistics converged to a point where the fraction of residues in the favored region was abnormally low (~80%) despite generally good geometry otherwise. To improve these statistics, Ramachandran restraints were introduced, but the default settings did not improve the models. *phenix.real_space_refine* offers two target functions, *emsley* and *oldfield*; the details of these implementations are discussed elsewhere (*Headd et al., 2012*). Both approaches rely on several empirically-determined weights which are currently not optimized on a per-refinement basis by *phenix.real_space_refine.*

To improve the Ramachandran statistics and geometry of the models, a two-parameter grid search of real space refinements was performed in which 3–5 macrocycles of global minimization and local grid search were performed in the presence of secondary structure restraints. First, a grid refinement search was performed for both target functions with around 1000 refinements each. For the *emsley* target function, *rama_weight* and *scale_allowed* were varied from 0.01 to 300; for the *oldfield* target function, a grid of refinements was performed over *plot_cutoff* values from 0.01 to 1.0 and *weight_scale* values from 0.1 to 300. Results were judged empirically and based primarily on a balance between $CC_{mask}$ and a minimal fraction of residues flagged by the program CaBLAM (*Richardson et al., 2018*) because focusing on the fraction of residues with favored $CC_{mask}$ and Ramachandran statistics alone often resulted in unrealistic models with serious problems (*Figure 12*, *Table 2*).

While optimal weights depended somewhat on the input model, several trends emerged. First, given the parameters supplied, the *oldfield* method was found to perform better than the *emsley* method. The *oldfield* potential consistently produced models with both good overall geometry and a Ramachandran fraction favored >98%; in addition, no reduction in $CC_{mask}$ was observed relative to the input model. Typically, *plot_cutoff* was on the order of 0.1–0.5, and *weight_scale* was on the order of 0–10. On the contrary, the *emsley* method failed to produce any models with a fraction favored >94%, and no models with acceptable geometry were identified with a fraction favored >92%. While bond length RMSDs were often adequate for *emsley* models with a fraction favored >92%, bond angle RMSDs were unacceptably large. Finally, these models fit the data more poorly than the best *oldfield* models, with $CC_{mask}$ generally a few percent worse for models with a fraction favored >90%. These results are likely method- (i.e., X-ray vs. electron), model-, and resolution-specific; scripts for performing this refinement protocol are provided.

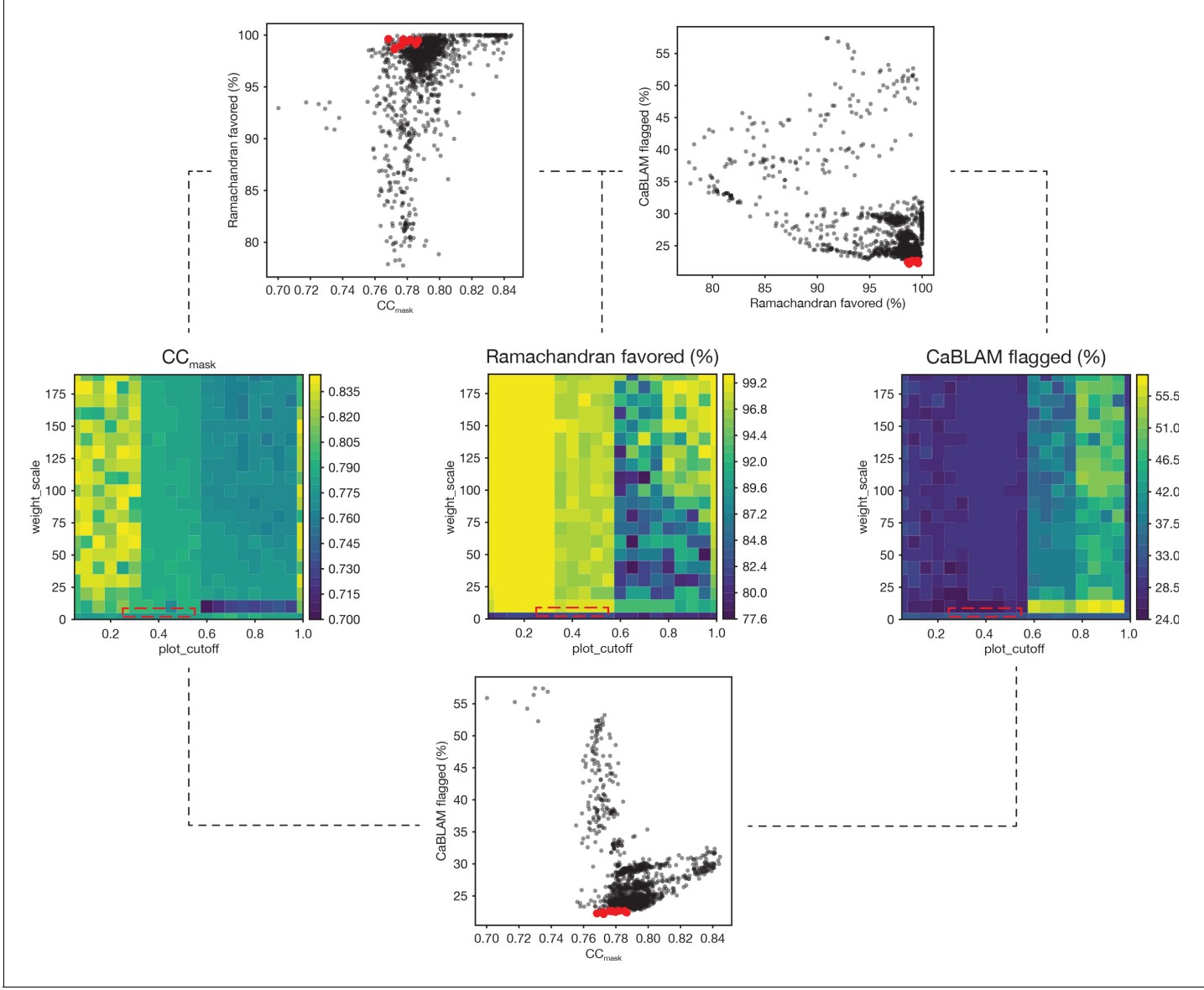

**Figure 12.** Summary of a 1,306-point refinement grid search. CaBLAM scores were used to guide the enforcement of Ramachandran restraints. At several points during the iterative refinement of the final FL-20S$_{focus}$ models, Ramachandran restraints were imposed using the Oldfield target function (*Oldfield, 2001*) as implemented in *phenix.real_space_refine* (*Adams et al., 2010*; *Headd et al., 2012*). A grid of refinements with different values of the parameters *weight_scale* and *plot_cutoff* were performed; the results are visualized as surfaces for three key diagnostic parameters—*CC$_{mask}$*, Ramachandran fraction favored, and the total fraction of residues flagged by CaBLAM analysis (*i.e.*, the total fraction of residues flagged as either outliers, disfavored, or severe) (*Richardson et al., 2018*). The value of *plot_cutoff* largely sorts results into three matching regions for each metric, while the value of *weight_scale* is less predictive. The ten refinements with the lowest CaBLAM scores were further examined, and one model was chosen for subsequent manual and automated refinement. The boxes (dashed lines) on the surface plots illustrate the regions from which these models were found in parameter space. Scatter plots reveal the relationships between CCmask, Ramachandran fraction favored, and CaBLAM fraction flagged; the top ten refinements as identified by CaBLAM are shown (red points).

DOI: https://doi.org/10.7554/eLife.38888.021

This approach produced models with improved Ramachandran statistics and geometry (*Tables 1* and *2*). Model quality was assessed using MolProbity (*Davis et al., 2007*). In general, this approach introduced additional cis and twisted peptides which were then resolved manually.

Rigid body refinement in *phenix.real_space_refine* was then used to place FL-20S$_{focus}$-1 and FL-20S$_{focus}$-2 models (*Zhao et al., 2015*) into unsharpened FL-20S-1 and FL-20S-2 density, respectively.

**Table 2.** Refinement statistics for the ten models out of 1306 with the lowest fraction of residues flagged by CaBLAM (see *Figure 12*). The model corresponding to *plot_cutoff* = 0.30 and *weight_scale* = 2.51 (row highlighted in red) was chosen for further rounds of refinement based on good overall geometry and a minimal number of twisted residues.

| Oldfield plot_cutoff | Oldfield weight_scale | CaBLAM flagged (%) | CC$_{mask}$ | Clashscore | Bonds RMSD (Å) | Angles RMSD (°) | Ramachandran outliers (%) | Ramachandran allowed (%) | Ramachandran favored (%) | Rotamer outliers (%) | cis-general (%) | twisted general (%) | Chirality RMSD | Dihedral RMSD (°) |
|---|---|---|---|---|---|---|---|---|---|---|---|---|---|---|
| 0.30 | 4.21 | 17.38 | 0.769 | 3.26 | 0.005 | 1.068 | 0 | 0.31 | 99.69 | 0.12 | 0 | 2.00 | 0.089 | 12.159 |
| 0.30 | 3.71 | 17.58 | 0.765 | 3.09 | 0.004 | 1.064 | 0 | 0.27 | 99.73 | 0.16 | 0 | 2.03 | 0.088 | 12.322 |
| 0.20 | 5.21 | 17.64 | 0.773 | 3 | 0.006 | 1.066 | 0 | 0.24 | 99.76 | 0.12 | 0 | 1.35 | 0.09 | 12.253 |
| 0.25 | 4.31 | 17.71 | 0.778 | 3.43 | 0.007 | 1.102 | 0 | 0.45 | 99.55 | 0.12 | 0 | 1.64 | 0.091 | 12.498 |
| 0.30 | 3.81 | 17.75 | 0.774 | 3.58 | 0.006 | 1.085 | 0 | 0.41 | 99.59 | 0.16 | 0 | 1.93 | 0.09 | 12.44 |
| 0.30 | 5.21 | 17.85 | 0.791 | 3.85 | 0.012 | 1.283 | 0 | 0.45 | 99.55 | 0.36 | 0 | 2.50 | 0.105 | 13.306 |
| 0.30 | 4.31 | 17.88 | 0.788 | 3.83 | 0.011 | 1.247 | 0 | 0.52 | 99.48 | 0.16 | 0 | 2.14 | 0.102 | 13.096 |
| 0.25 | 4.91 | 17.91 | 0.783 | 3.73 | 0.008 | 1.14 | 0 | 0.38 | 99.62 | 0.16 | 0 | 1.82 | 0.094 | 12.795 |
| 0.25 | 4.21 | 17.91 | 0.785 | 3.66 | 0.01 | 1.174 | 0 | 0.45 | 99.55 | 0.08 | 0 | 1.64 | 0.096 | 12.712 |
| 0.20 | 7.11 | 17.91 | 0.776 | 3.45 | 0.006 | 1.095 | 0 | 0.1 | 99.9 | 0.16 | 0 | 1.75 | 0.091 | 12.258 |

DOI: https://doi.org/10.7554/eLife.38888.022

In both FL-20S-1 and FL-20S-2 classes, a single copy of the crystal structure of the neuronal SNARE complex (PDB 1SFC) (*Sutton et al., 1998*), two copies of an αSNAP homology model derived from a crystal structure of Sec17 (*Rice and Brunger, 1999*; *Zhao et al., 2015*), and four copies of the N domain (PDB 1QCS) (*Yu et al., 1999*) were placed, and rigid body minimization was performed against unsharpened maps. The 16 residue N-terminal end of SNAP-25A was then manually extended to join the neuronal SNARE complex. Finally, the nucleotide states of the protomer A and E D1 domains are ambiguous; corresponding densities were modeled as ADP in the deposited coordinate files due to a lack of strong γ-phosphate signal.

Figures were prepared either with PyMOL (The PyMOL Molecular Graphics System, version 2.0.2 Schrödinger, LLC.) or UCSF Chimera (version 1.12). Chimera is developed by the Resource for Biocomputing, Visualization, and Informatics at the University of California, San Francisco (supported by NIGMS P41-GM103311). Electrostatic surfaces were calculated using APBS (*Baker et al., 2001*) as implemented in UCSF Chimera. Volume segmentation was performed using Segger 1.9.4 as implemented in UCSF Chimera (*Pintilie et al., 2010*).

## Fluorescence dequenching assay of NSF-mediated disassembly of SNARE complex

The details of the fluorescence dequenching assay have been previously described (*Choi et al., 2018*; *Cipriano et al., 2013*; *Vivona et al., 2013*; *Zhao et al., 2015*). Purified soluble rat neuronal SNARE complex was mixed with a 20-fold molar excess of thiol-reactive Oregon Green 488 maleimide dye and nutated overnight at 4°C. The next day, excess dye was removed by buffer exchange and protein was concentrated, flash frozen, and stored at −80°C.

Fluorescence dequenching assays were carried out using a FlexStation II 384-well plate reader (Molecular Devices) with a final reaction volume of 60 μL. All conditions were assayed at the same time with five or six replicates for each. All reactions were carried out in reaction buffer composed of 20 mM Tris pH 8, 100 mM NaCl, 2 mM ATP, and 0.5 mM TCEP. The assay plate was first prepared by adding 50 μL of 60 nM NSF (either wild-type, Y294A mutant, or Y294L mutant), 2.4 μM αSNAP, 480 nM OG-labeled wild type or SNARE$_{\Delta16}$ complex in reaction buffer to each well. Then, 100 μL of reaction buffer (as a control) or ATP hydrolysis initiation buffer (reaction buffer supplemented with 24 μM MgCl$_2$) was added to appropriate wells of the compound plate. A program was run to monitor fluorescence dequenching. Excitation was performed at 485 nm, and emission was monitored at 525 nm with a dichroic mirror set at 515 nm. First, fluorescence intensity was monitored prior to reaction initiation for 60 s. Then, 10 μL of compound plate solution was transferred to each well of the assay plate (diluting all protein 5/6-fold and MgCl$_2$ concentrations six-fold) and triturated once to initiate the disassembly reaction. Final protein concentrations following mixing were 50 nM NSF, 2.0 μM αSNAP, and 400 nM OG-labeled wild type SNARE or SNARE$_{\Delta16}$ complex. The final MgCl$_2$ concentration was 4 mM. Dequenching was then monitored for 30 min at 20°C. Linear regression over the first several minutes was performed to estimate relative rates of disassembly. Statistical significance of changes to disassembly rates was assessed using the two-tailed Student's t-test.

## Intrinsic NSF ATPase activity assay

The intrinsic rate of ATP hydrolysis by NSF was measured using a photometric assay that monitors the conversion of NADH into NAD$^+$ through pyruvate kinase and lactate dehydrogenase at 340 nm; the details of this assay have been published previously (*Nørby, 1988*; *Vivona et al., 2013*). Reactions were performed simultaneously in a Cary 100 UV-Vis spectrophotometer maintained at 37°C. First, a solution containing 24 nM NSF (wild-type, Y294A, or Y294L) was prepared in reaction buffer composed of 50 mM Tris pH 8.0, 4 mM MgSO$_4$, 2 mM ATP, 3 mM phosphoenolpyruvate, and 2.1% v/v pyruvate kinase/lactic dehydrogenase enzymes from rabbit muscle (Sigma P0294). Reference solutions with regeneration components only with and without NADH were also prepared. Solutions were added to cuvettes and brought to 37°C, and absorbance was monitored at 340 nm. The rate of NADH loss was calculated from the slope of the absorbance trace and was directly converted to the equivalent rate of ATP hydrolysis.

## Acknowledgements

We wish to thank Doeke Hekstra and Susan Wu for critical reading of this manuscript. This work was supported by a grant of the National Institutes of Health to ATB (R37MH63105) and by a postdoctoral fellowship from the Helen Hay Whitney Foundation supported by the Howard Hughes Medical Institute awarded to KIW.

## Additional information

### Competing interests

Axel T Brunger: Reviewing editor, *eLife*. The other authors declare that no competing interests exist.

### Funding

| Funder | Grant reference number | Author |
| --- | --- | --- |
| Howard Hughes Medical Institute | | Axel T Brunger |
| National Institutes of Health | R37MH63105 | Axel T Brunger |
| Helen Hay Whitney Foundation | | K Ian White |

The funders had no role in study design, data collection and interpretation, or the decision to submit the work for publication.

### Author contributions

K Ian White, Conceptualization, Software, Formal analysis, Validation, Investigation, Visualization, Methodology, Writing—original draft, Writing—review and editing; Minglei Zhao, Conceptualization, Formal analysis, Investigation, Writing—review and editing; Ucheor B Choi, Validation, Investigation; Richard A Pfuetzner, Resources, Performed preparations of protein samples; Axel T Brunger, Conceptualization, Supervision, Funding acquisition, Writing—original draft, Project administration, Writing—review and editing

### Author ORCIDs

K Ian White (iD) http://orcid.org/0000-0001-8182-3655
Minglei Zhao (iD) http://orcid.org/0000-0001-5832-6060
Ucheor B Choi (iD) http://orcid.org/0000-0003-1541-2967
Richard A Pfuetzner (iD) http://orcid.org/0000-0002-1741-2330
Axel T Brunger (iD) http://orcid.org/0000-0001-5121-2036

### Decision letter and Author response

Decision letter https://doi.org/10.7554/eLife.38888.033
Author response https://doi.org/10.7554/eLife.38888.034

## Additional files

### Supplementary files

• Transparent reporting form
DOI: https://doi.org/10.7554/eLife.38888.023

### Data availability

The coordinates and corresponding EM density maps have been deposited in the PDB and EMDB, respectively.

The following datasets were generated:

| | Database, license, and accessibility |
| --- | --- |

| Author(s) | Year | Dataset title | Dataset URL | information |
|---|---|---|---|---|
| White KI, Zhao M, Brunger AT | 2018 | The D1 and D2 domain rings of NSF engaging the SNAP-25 N-terminus within the 20S supercomplex (focused refinement on D1/D2 rings, class 1) | http://www.rcsb.org/structure/6MDO | Publicly available at the RCSB Protein Data Bank (accession no: 6MDO) |
| White KI, Zhao M, Brunger AT | 2018 | The 20S supercomplex engaging the SNAP-25 N-terminus (class 1) | http://www.rcsb.org/structure/6MDM | Publicly available at the RCSB Protein Data Bank (accession no: 6MDM) |
| White KI, Zhao M, Brunger AT | 2018 | The 20S supercomplex engaging the SNAP-25 N-terminus (class 2) | http://www.rcsb.org/structure/6MDN | Publicly available at the RCSB Protein Data Bank (accession no: 6MDN) |
| White KI, Zhao M, Brunger AT | 2018 | The D1 and D2 domain rings of NSF engaging the SNAP-25 N-terminus within the 20S supercomplex (focused refinement on D1/D2 rings, class 2) | http://www.rcsb.org/structure/6MDP | Publicly available at the RCSB Protein Data Bank (accession no: 6MDP) |

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
