## [Decision Letter]

Thank you for your submission of your manuscript entitled "Structural principles of SNARE complex recognition by the AAA+ protein NSF." Your manuscript was reviewed by three independent reviewers, who all agreed that the work represented a significant advance in this field, especially as related to the discovery of the organization of the SNARE helices relative to the NSF D1 ring and potential mechanistic insights into SNARE disassembly. Sriram Subramaniam served as the Reviewing Editor for the manuscript. The following individual involved in review of your submission has agreed to reveal his identity: Gabriel C Lander (Reviewer #3).

The reviewers have discussed the reviews with one another and the Reviewing Editor has drafted this decision to help you prepare a revised submission.

A number of concerns emerged from the review, as noted below. These concerns would need to be addressed in a revised version in order for this manuscript to be acceptable for publication in *eLife*.

1) A principal concern is with the assignment of nucleotide states in the structure. The authors state that the generalized "hand-over-hand" mechanism of translocation holds true for NSF, but the assigned nucleotide states at either side of the "seam" subunit (F) are not consistent with this mechanism. The generalized model posits that subunit F, which has released substrate and nucleotide, would be the next protomer to bind ATP and assume the uppermost position in the staircase (represented by E). Thus, the D:E pocket should have a much higher affinity for ATP than ADP, so it is unclear why a contaminating ADP would be bound at this pocket. The authors mention that the nucleotide density at the D:E pocket corresponds to an ADP or mixed state nucleotide, but it's unclear from the orientation presented in Figure 11C that this is the case. Have the authors checked to see if an ATP can be accommodated within the density? It looks as though R385 and 385 of subunit D are in close proximity to the phosphates, which would be more consistent with an ATP, rather than an ADP.

2) Second, according to the hand-over-hand mechanism, subunits E, D, C, B should contain ATP, with their pore loops closely associated with substrate. Hydrolysis would occur in the lower-most subunit, in the F:A pocket, and the substrate interactions are lessened as a result. Even though ATP hydrolysis has been prevented in the purification of this complex, it is surprising to see an ATP in this pocket, when other (hydrolysis deficient and analog-bound) AAA+ ATPases have observed ADP in this corresponding site. How confident are the authors of the nucleotide states in these pockets? An expanded discussion concerning these deviations from previously observed nucleotide states, and hydrolysis mechanism is necessary to clarify this point.

3) The authors suggest that the map has density for the SNAP-25 linker, but the presentation in Figure 7 is unclear. This requires either additional labels or perhaps another representation that captures this result.

4) Is there a clear correspondence between the states of the 20S observed in this study (e.g. in Figure 3C) and the four states observed in the authors' earlier (2015) study of 20S with AMPPNP?

5) The authors also state, qualitatively, that R385 and 385 of subunit E (in the E:F pocket) are "somewhat retracted". What exactly does this mean? Is it supported credibly by the cryo-EM density?

6) The EM density for the ATP binding pockets in Figure 11 should be shown with the inclusion of the surrounding EM density to give a better sense of the quality of the density. The authors have "zoned" the EM density surrounding the nucleotide, a strategy that can be misleading in conveying the actual local quality of the map. Density for all 6 nucleotide pockets should be shown.

7) It is not clear why focused analysis of the spire region was not done. This would enable the authors to expand on the αSNAP-SNARE interactions, which are limited to superficial observations about complementary electrostatics of the surfaces. If such analyses were indeed performed and did not yield any improvements then this should be mentioned in the text. Further, the authors indicate that D1 and D2 rings within the two FL-20S_focus_ classes are in the same conformations, but slightly separated from one another to differing extents. If the particles in these classes are combined and reprocessed with masks only around the D1 or D2 rings, are these regions improved in resolution?

8) In the fourth paragraph of the subsection “Mechanistic implications of the FL-20S structure”, the secondary pore loop is discussed as having a "guidance" role in substrate translocation, and the text references to Figure 9A. However, pore loop 2 isn't clearly depicted in Figure 9, or indeed in any of the figures. Since this loop was not observed in prior structures, pore loop 2 should be depicted in a figure. If there is any relevant mutagenesis data for this pore loop, it should be mentioned.

---

## [Author Response]

A number of concerns emerged from the review, as noted below. These concerns would need to be addressed in a revised version in order for this manuscript to be acceptable for publication in eLife.1) A principal concern is with the assignment of nucleotide states in the structure. The authors state that the generalized "hand-over-hand" mechanism of translocation holds true for NSF, but the assigned nucleotide states at either side of the "seam" subunit (F) are not consistent with this mechanism. The generalized model posits that subunit F, which has released substrate and nucleotide, would be the next protomer to bind ATP and assume the uppermost position in the staircase (represented by E). Thus, the D:E pocket should have a much higher affinity for ATP than ADP, so it is unclear why a contaminating ADP would be bound at this pocket. The authors mention that the nucleotide density at the D:E pocket corresponds to an ADP or mixed state nucleotide, but it's unclear from the orientation presented in Figure 11C that this is the case. Have the authors checked to see if an ATP can be accommodated within the density? It looks as though R385 and 385 of subunit D are in close proximity to the phosphates, which would be more consistent with an ATP, rather than an ADP.2) Second, according to the hand-over-hand mechanism, subunits E, D, C, B should contain ATP, with their pore loops closely associated with substrate. Hydrolysis would occur in the lower-most subunit, in the F:A pocket, and the substrate interactions are lessened as a result. Even though ATP hydrolysis has been prevented in the purification of this complex, it is surprising to see an ATP in this pocket, when other (hydrolysis deficient and analog-bound) AAA+ ATPases have observed ADP in this corresponding site. How confident are the authors of the nucleotide states in these pockets? An expanded discussion concerning these deviations from previously observed nucleotide states, and hydrolysis mechanism is necessary to clarify this point.

We thank the reviewers for these two suggestions which prompted us to re-examine the FL-20S_focus_-1/2 maps as well as the structures of other AAA+ proteins with bound substrate. Regarding our maps, we previously noted that the density for the D1 domain of the E protomer is ambiguous. Upon closer examination, the density is also somewhat ambiguous for the A protomer, although less so than for the E protomer. In our original model, we fit an ATP into the A protomer, but the γ-phosphate group density is weak. In both cases this could arise for a number of non-mutually exclusive reasons; there could be contaminating amounts of ADP in the sample buffer and/or the lower resolution of the maps for the nucleotide binding regions of protomers A and E (Figure 1—figure supplement 3) lead to weak density that is challenging to interpret as a unique entity. In light of these limitations, we modeled the nucleotides at the A and E positions as ADP in the deposited coordinates, and have noted that they are ambiguous in both the text and in figures (as A[D/T]P).

More generally, we propose a new hydrolysis-free mechanism for initial substrate engagement in the manuscript based our findings as well as the findings of others for related AAA+ proteins. The current knowledge of AAA+-substrate engagement is based on static structures of what is likely the initial state of the each of these systems; although the hand-over-hand model is an attractive model based on these findings, direct validation will require studies of intermediate states in all of these systems, including our own. We have revised and expanded the discussion of this model in our manuscript.

3) The authors suggest that the map has density for the SNAP-25 linker, but the presentation in Figure 7 is unclear. This requires either additional labels or perhaps another representation that captures this result.

We apologize for the lack of labeling in Figure 7. We have introduced labels that will hopefully clarify our point.

4) Is there a clear correspondence between the states of the 20S observed in this study (e.g. in Figure 3C) and the four states observed in the authors' earlier (2015) study of 20S with AMPPNP?

Yes. The FL-20S-1 and FL-20S_focus_-2 structures are most similar to the second class from the previous study, and the FL20S-2 and FL-20S_focus_-1 structures are most similar to both class 3a and the structure of the V7-20S complex from the previous study. Despite these similarities, the new FL classes each have a pair of adjacent N-domains associated with each of two adjacent αSNAP molecules, whereas the previous structures had four total αSNAP molecules engaged by a more heterogeneous N-domain engagement pattern (depending on the specific class). We have revised the text and included an additional supplemental figure (Figure 3—figure supplement 1) to make this clearer.

5) The authors also state, qualitatively, that R385 and 385 of subunit E (in the E:F pocket) are "somewhat retracted". What exactly does this mean? Is it supported credibly by the cryo-EM density?

While there are differences in EM density in this region, the quality for subunit E is quite poor. In light of this, and the fact that there is no obvious density for subunit F nucleotide, we have removed this statement about the subunit E arginine fingers.

6) The EM density for the ATP binding pockets in Figure 11 should be shown with the inclusion of the surrounding EM density to give a better sense of the quality of the density. The authors have "zoned" the EM density surrounding the nucleotide, a strategy that can be misleading in conveying the actual local quality of the map. Density for all 6 nucleotide pockets should be shown.

We agree and now show all 6 nucleotide pockets in Figure 11, and include a new supplementary figure (Figure 11—figure supplement 1) that shows non-segmented density maps for both nucleotide and protein at different contours for one of these pockets. We note that we did not include the sidechain densities in the main Figure 11 for the sake of clarity.

7) It is not clear why focused analysis of the spire region was not done. This would enable the authors to expand on the αSNAP-SNARE interactions, which are limited to superficial observations about complementary electrostatics of the surfaces. If such analyses were indeed performed and did not yield any improvements then this should be mentioned in the text. Further, the authors indicate that D1 and D2 rings within the two FL-20S_focus_ classes are in the same conformations, but slightly separated from one another to differing extents. If the particles in these classes are combined and reprocessed with masks only around the D1 or D2 rings, are these regions improved in resolution?

We indeed tried this approach to improve the resolution of the spire region, but unfortunately were unsuccessful. We speculate that the focused approach does not work well for the spire due to its relatively small size compared to the NSF hexamer. With respect to the two focused classes FL-20S_focus_-1 and FL-20S_focus_-2 we note that they were obtained by using a combined mask around the D1 and D2 rings only (i.e., the mask excluded the spire, including the N-terminal NSF domains). Weak unmodeled spire density (for the SNARE complex, αSNAP molecules and the N-terminal domain of NSF) supports the separation of the two focused classes based on N domain and αSNAP configuration. These regions are subtly coupled to nearby D1 domains (particularly the unmodeled protomer F D1 domain).

8) In the fourth paragraph of the subsection “Mechanistic implications of the FL-20S structure”, the secondary pore loop is discussed as having a "guidance" role in substrate translocation, and the text references to Figure 9A. However, pore loop 2 isn't clearly depicted in Figure 9, or indeed in any of the figures. Since this loop was not observed in prior structures, pore loop 2 should be depicted in a figure. If there is any relevant mutagenesis data for this pore loop, it should be mentioned.

We thank the reviewers for the suggestion and have included a new supplement to Figure 9 to more clearly show pore loop 2. Unfortunately, there are no mutagenesis data available for this loop.